# TOWARDS VISUAL TEXT GROUNDING OF MULTIMODAL LARGE LANGUAGE MODEL

## ABSTRACT

Despite the existing evolution of Multimodal Large Language Models (MLLMs), a non-negligible limitation remains in their struggle with visual text grounding, especially in text-rich images of documents. Document images, such as scanned forms and infographics, highlight critical challenges due to their complex layouts and textual content. However, current benchmarks do not fully address these challenges, as they mostly focus on visual grounding on natural images, rather than text-rich document images. Thus, to bridge this gap, we introduce **TRIG**, a novel task with a newly designed instruction dataset for benchmarking and improving the **T**ext-**R**ich **I**mage **G**rounding capabilities of MLLMs in document question-answering. Specifically, we propose an OCR-LLM-human interaction pipeline to create 800 manually annotated question-answer pairs as a benchmark and a large-scale training set of 90k synthetic data based on four diverse datasets. A comprehensive evaluation of various MLLMs on our proposed benchmark exposes substantial limitations in their grounding capability on text-rich images. In addition, we propose two simple and effective TRIG methods based on general instruction tuning and plug-and-play efficient embedding, respectively. By finetuning MLLMs on our synthetic dataset, they promise to improve spatial reasoning and grounding capabilities. The data and code are available at https://anonymous.4open.science/r/TRIG-Bench-5BD8/

## 1 INTRODUCTION

Despite the remarkable advancements in LLMs and MLLMs, the trustworthiness of their generated outputs remains a critical concern (Liu et al., 2024c; Sun et al., 2024). While these models can produce fluent and coherent responses, they often lack grounding capability, which can lead to potential hallucinations (Ji et al., 2023; Rawte et al., 2023; Liu et al., 2024a). Grounding capability is defined as the model's ability to accurately localize relevant regions in the visual content based on the provided semantic description (Nagaraja et al., 2016; Luo & Shakhnarovich, 2017; Yu et al., 2017; Kamath et al., 2021; You et al., 2024). This capability is essential for ensuring that the responses are accurate and verifiable.

Existing grounding efforts in MLLMs mainly focus on natural images, where the task involves associating textual descriptions with corresponding visual elements, such as objects or scenes(Plummer et al., 2015; Chen et al., 2023b; Zhang et al., 2023a; 2024b; Peng et al., 2024; You et al., 2024). However, there is a significant gap in the literature when it comes to visual text grounding in text-rich document images. Document images, such as scanned forms, charts, and complex posters, present unique challenges that differ markedly from those found in natural images (Mathew et al., 2021; Masry et al., 2022; Mathew et al., 2022; Zhang et al., 2024a). They often feature a mix of textual and graphical elements and require precise localization and understanding of content, especially textual content. An example is shown in Figure 1, for the given text-rich document image, we expect LLMs to not only generate the answer alone but also to provide the corresponding grounded bounding boxes that can support its answer, which requires deeper spatial understanding and reasoning capabilities, and sometimes better instruction-following abilities to correctly provide the grounding information in the desired formats.

Despite its importance, there is no established task specifically designed to evaluate the visual text grounding capabilities of MLLMs on Text-Rich Document Question-Answering tasks and no well-

formulated corresponding instruction datasets and baseline methods for the purpose. The absence of such a well-formulated task limits the ability to systematically assess and improve the performance of different models in this domain. Thus to bridge this gap, we introduce **TRIG**, a novel **T**ext-**R**ich **I**mage **G**rounding task with instruction set for document QA grounding, along with its corresponding benchmark notated as **TRIG-Bench**. TRIG-Bench consists of 800 question-answer pairs manually collected from DocVQA (Mathew et al., 2021), ChartQA (Masry et al., 2022), InfographicsVQA (Mathew et al., 2022), and TRINS datasets (Zhang et al., 2024a), along with human-inspected ground-truth bounding boxes that support the answer to the corresponding question. The TRIG instruction dataset contains approximately 90k training instances generated and verified by the GPT4o model from the above-mentioned data sources. They provide a standardized framework for evaluating the ability of MLLMs to accurately ground their responses to document-related visual questions.

For text-rich document images, we mainly focus on the visual texts on them as the main grounding target. Considering the supreme performance of modern OCR models and the promising reasoning ability of current LLMs like GPT4o, an OCR-LLM-Human interaction pipeline is proposed for the benchmark construction. Specifically, for every given VQA pair, we first utilize PaddleOCR to detect and recognize all the texts. Given all these OCR results, LLMs are asked to judge which bounding boxes support the answer to the corresponding question, followed by another LLM evaluating the correctness of the chosen grounded bounding boxes. The resulting data has already been of good quality, as it is generated and verified by the powerful GPT4o model. Next, more human participants will manually inspect the generated results and add the correct ones to build our benchmark data.

In addition, two types of methods are further proposed, including the general **instruction-tuning-based method** and a novel **embedding-based method**, as potential baselines for TRIG. Specifically, for the instruction-tuning-based method, following the implementation of modern MLLM

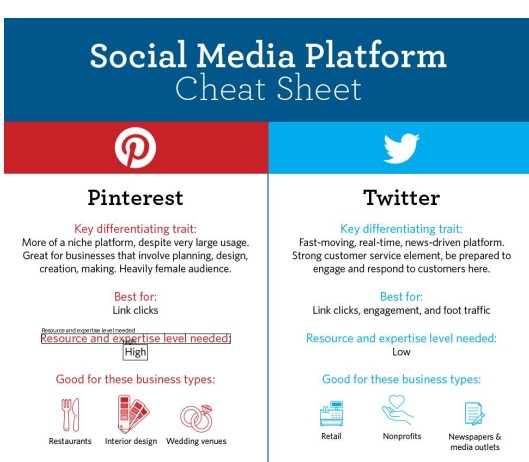

Figure 1: An example from InfograhicsVQA. **Question:** What is the resource and expertise level needed for Pinterest? **Answer:** high. The LLM is expected to generate the answer together with the corresponding grounded bounding boxes that can support its answer, which requires deeper spatial understanding and reasoning, and sometimes instruction-following abilities.

instruction tuning (Liu et al., 2023b), we train an MLLM with supreme document grounding capability. Moreover, we further propose an efficient plug-and-play embedding-based method, whose inference efficiency is highly promising due to the unnecessity of the iterative token generation process, which largely speeds up the grounding process, though with lower performance scores. Our main contributions:

- We introduce a novel task, text-rich document image grounding (**TRIG**), its corresponding benchmark **TRIG-Bench**, and an instruction tuning dataset. The dataset and benchmark are the first of their kind and provide a standardized framework for evaluating MLLMs in this domain, filling a critical gap in existing research.

- We propose two types of methods to tackle the challenging visual text grounding task, including the straightforward instruction-tuning-based method and a novel efficient embedding-based method, which not only provides the baseline performances, but also provides more insight into document grounding.

- We perform a comprehensive evaluation of a range of existing MLLMs on our new benchmark. Our analysis provides a deeper understanding of the limitations and constraints of current MLLMs: Although most of the current MLLMs perform well on well-defined tasks, they lack the capability to follow customized and complex tasks that require deep understanding and reasoning capabilities.

## 2 TRIG: TEXT-RICH IMAGE GROUNDING

Text-Rich Image Grounding (TRIG) is specifically designed for MLLM-based grounding on text-rich image QA tasks, which is still an underexplored topic. We aim to build the corresponding instruction dataset, benchmark, and evaluation protocols for this task. Specifically, given a text-rich document image and a question, MLLMs are instructed to generate the bounding boxes of the area that supports answers to the given question. As shown in Figure 1, given the image and the question, the models should generate the bounding boxes directly supporting the final answer.

To ensure the diversity of TRIG, four existing text-rich image datasets are chosen as our data sources, covering a wide range of document image types[1], including DocVQA (Mathew et al., 2021), ChartQA (Masry et al., 2022), InfographicVQA (Mathew et al., 2022), and TRINS (Zhang et al., 2024a). Examples and detailed illustrations can be found in the supplementary material. The following subsections illustrate the training&benchmark data construction pipeline and evaluation protocols of TRIG.

### 2.1 DATA CONSTRUCTION

Our OCR-LLM-Human interactive pipeline for training data and benchmark data generation is shown in Figure 2.

**Step 1 Preprocessing:** PaddleOCR[2] is utilized to obtain the initial OCR information for its simplicity and promising performance.

**Step 2 Generation:** After obtaining the OCR information, a critical issue is how to transmit the OCR information to the LLMs.

The common method in the literature is wrapping all the obtained OCR information around the prompt (Zhang et al., 2023b; Wang et al., 2023b), however, we observe a severe misalignment between the text information in the prompt and the visual information in the image, leading to unpromising results. Thus, to further align the visual and text information provided in different genres, we innovatively assign every OCR bounding box an identical index, draw every bounding box with its index on the original images, and simultaneously provide all the OCR information with the index in the prompt. By utilizing this strategy, the LLM is able to better align each visual element with its detailed information. Along with the original question and ground truth answer, the LLM is prompted to select the bounding boxes that can support the given answer to the corresponding question. **Step 3 Correction:** Then another Reflection & Rectification module is introduced to examine the correctness of selected bounding boxes (Pan et al., 2023; Huang et al., 2023). In addition to all the previous information used in Step 2, the previously generated bounding box indices are also provided to the LLM, prompting it to judge if the selected grounded bounding boxes can adequately lead to the given answer. If the judging result is "correct", this sample will be kept if it is from the training set, or sent to the next step if it is used for building a benchmark. Otherwise, it will be sent back to the previous step until reaching the

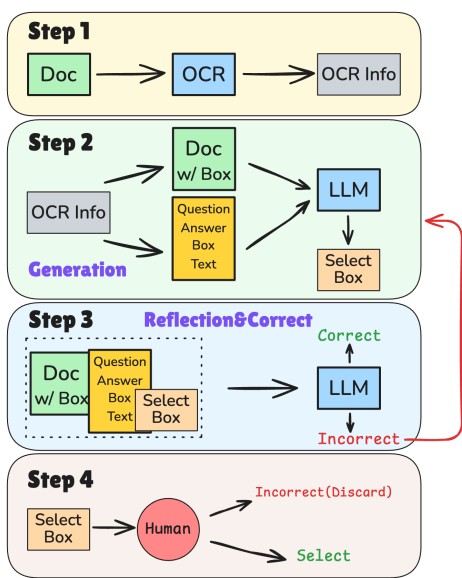

Figure 2: Main Constriction Pipeline. The pipeline contains 4 steps: Preprocessing, Generation, Correction, and Human Evaluation. The benchmark data will go through all of these 4 steps, and the training data will go through the previous 3 steps.

maximum rounds and being discarded. The resulting data after the previous 3 steps has already been of good quality, as it is generated and verified by the powerful LLM.

---

[1]The detailed description of data sources can be found in Appendix C, and more benchmark data examples can be found in Appendix I.

[2]https://github.com/PaddlePaddle/PaddleOCR

**Step 4 Human Evaluation:** The sample sent to step 4 has been evaluated and rectified by LLM, however, its quality might still have discrepancies with the requirements of being a benchmark. Thus, to ensure the correctness of our benchmark data, further human inspection is conducted. Specifically, two human participants are invited for the human evaluation. Only those samples that are agreed to be correct for both participants can be kept in our benchmark; otherwise, they will be discarded. Finally, for each dataset, 200 QA pairs are manually selected as the ground truth of this benchmark.

Examples from our benchmark and data statistics can be found in the supplementary material, including (1) the average question, answer, and OCR text word counts; (2) the number of bonding boxes provided by OCR models, selected as the ground truth and their ratio; (3) the area of bonding boxes provided by OCR models, selected as the ground truth and their ratio. From these statistics, the characteristics of each dataset can be illustrated, which showcases the variety of our data components and further provides a clear understanding of evaluation.

## 2.2 Evaluation Settings

To simulate various real-world scenarios and test models' capability for grounding at different levels, two different evaluation settings are provided that are compatible with this benchmark. All the prompts used for evaluation can be found in the supplementary material.

**OCR-free Grounding (Setting 1)**  In this setting, only the document image and corresponding question are provided, together with the specific instruction that describes the task requirement and defines the desired format. It represents the hardest level of MLLMs' grounding capabilities as it requires MLLMs to answer the question and simultaneously generate the corresponding grounded bounding boxes from scratch. In addition to measuring MLLMs' grounding capabilities, it also measures their instruction-following abilities. As a complex, customized task, the instructions for generating grounded bounding boxes have never been seen by MLLMs during training, and it require deep reasoning capability to correctly follow. **Pixel-level IoU** (Intersection over Union) is utilized as the evaluation metric. It calculates the overlap between pixels in the predicted bounding boxes and ground truth bounding boxes, normalized by the total number of unique pixels. It is commonly used in image segmentation tasks to assess the accuracy of pixel-wise predictions.

**OCR-based Grounding (Setting 2)**  Although the previous setting best aligns with our goal, it is still difficult for most existing models, from proprietary to open-source models. Thus, the setting of OCR-based Grounding is proposed, where an additional OCR model is utilized to facilitate the generation of grounded bounding boxes. Specifically, all the bounding box coordinates and corresponding text content obtained from the OCR model will be wrapped into the prompt for MLLMs as additional information. Under these circumstances, the bounding box generation task is converted into an easier bounding box selection task, in which MLLMs do not need to generate coordinates from scratch but only need to select the proper ones given in the prompt. It is worth noting that this evaluation setting still measures MLLMs' visual text grounding capability as they have to align the given OCR information to the image information.

In this setting, the IoU score at the instance level, precision, recall, and F1 score are evaluation metrics. **Instance-level IoU** measures the overlap between the selected grounded bounding boxes and the ground truth bounding boxes, normalized by the total number of unique elements. It evaluates how well the selected bounding boxes match the ground truth ones. **Precision** measures the proportion of correctly selected elements out of all elements selected by the model. It reflects the accuracy of the model's positive predictions. **Recall** measures the proportion of correctly selected elements out of all actual ground truth elements. It indicates the model's ability to capture all relevant elements in the ground truth. **F1 Score** is the harmonic mean of precision and recall, providing a balanced measure that accounts for both false positives and false negatives.

## 3 Proposed Methods

Along with the benchmark and corresponding training data, we propose two baseline methods for this task: (1) the instruction-tuning-based (Liu et al., 2023b; Zhang et al., 2023b) method (we use the term *instruction-based* for simplicity), which treats this grounding task as an instruction-following

task by iterative next token prediction, and (2) the efficient embedding-based method that directly finds the image patches that have the greatest similarities with the input text embeddings.

The **instruction-based method** formulates the generation of bounding boxes into a token-prediction framework, in which the inputs include a curated prompt $x$ and an image, and the output will be a sequence of tokens as the response $y$. For simplicity, we denote each data sample as a triplet (Image, $x$, $y$) and let $p_\theta(\cdot)$ denote the LLMs to be trained, with parameters $\theta$. In this type of method, the bounding box coordinates are also represented by sets of tokens. For simplicity, we denote the text string that represents each bounding box as $b_i$ for the $i$th bounding box. In the instruction tuning setting, $p_\theta$ is typically fine-tuned by minimizing the next-token-prediction loss on each data (Image, $x$, $y$):

Different from the previous method, the core of the **embedding-based method** is the corresponding embeddings of patches of images and text tokens. We denote the embeddings of image patches as $H^{\text{Image}} \in \mathbb{R}^{l_{\text{Image}} \times D}$, where $l_{\text{Image}}$ is the number of image patches, and $D$ is the dimension of embeddings, and we denote the embeddings of text tokens as $H^{\text{Text}} \in \mathbb{R}^{l_{\text{Text}} \times D}$, where $l_{\text{Text}}$ is the number of text tokens. $H_i^{\text{Image}}$ and $H_i^{\text{Text}}$ represents the $i$th embedding of the image patches or tokens, respectively.

### 3.1 INSTRUCTION-BASED METHOD

For the **OCR-free Grounding** setting, i.e., the evaluation setting 1, the input prompt contains only the original question and the instruction guiding LLMs to generate formatted grounded bounding boxes, which can be simply noted as $x$. The output of this setting includes the answer $y$ to the question concatenated with the texts of corresponding supportive bounding boxes, which can be denoted as $y'$. Thus, the loss function in this setting can be denoted as:

$$L_\theta = -\sum_{j=1}^{l} \log p_\theta \left( y'_j | \text{Image}, x, y'_{<j} \right), y' = [y; b_1; ..; b_G] \tag{1}$$

where $G$ is the number of golden grounded bounding boxes, and $[;]$ represents the concatenation.

For the **OCR-based Grounding** setting, i.e., the evaluation setting 2, all the bounding boxes generated by the OCR module will be provided in the input prompt, thus, the new input $x'$ in this setting will be the concatenation of the original question, the instruction guiding LLMs to select the supportive bounding boxes, and all the bounding boxes. The output of this setting is identical to the previous setting, which includes the answer $y$ to the question concatenated with the texts of corresponding supportive bounding boxes, which can be denoted as $y'$. Thus, the loss function in this setting can be denoted as:

$$L_\theta = -\sum_{j=1}^{l} \log p_\theta \left( y'_j \mid \text{Image}, x', y'_{<j} \right),$$
$$x' = [x; b_1; \ldots; b_N], \qquad y' = [y; b'_1; \ldots; b'_G]. \tag{2}$$

Where $N$ is the number of total bounding boxes generated by the OCR system, $G$ is the number of golden grounded bounding boxes, and $b'_1$ represents the first selected golden bounding box, which might be different from $b_1$.

In this type of method, a regular expression will first be utilized to extract the bounding box coordinates before calculating the evaluation scores.

### 3.2 EMBEDDING-BASED METHOD

In addition to the above instruction-based method, motivated by ColPali (Faysse et al., 2024), an embedding-based method is also provided. For instruction-based methods, the instruction-following capability of the LLMs severely affects the final performance, as shown in our experimental results. Most of the existing LLMs are not capable of following complex user-customized instructions like ours, leading to unsatisfactory performances. However, the embedding-based method disentangles the grounding capability from the instruction-following capability and makes this process much more efficient than continuous next-token prediction. Thus, it can serve as a plug-and-play module that is adaptive to MLLMs without grounding capability.

**Training**   In the embedding-based method, we fine-tune an MLLM as an encoder and estimate the grounding area using image-text embedding similarities. The text part is the concatenation of the question and the ground-truth answer. To maximize the image-text similarity within the grounding box, we apply a binary mask $\mathbf{M}$ to the patch embeddings, where $M_i = 1$ for patches overlapping with the grounding box and $M_i = 0$ otherwise. The training objective is based on a modified contextualized late interaction (col) score (Khattab & Zaharia, 2020), defined on the interaction matrix $\mathbf{X}$, where $X_{i,j} = H_i^{\text{Text}} \cdot (H_i^{\text{Image}})^T$. The masked col-score is then defined as: $s(\mathbf{X}, \mathbf{M}) = \sum_{i=1}^n \max_{\{j:M_j=1\}} X_{i,j}$. Following (Faysse et al., 2024), we use the hardest negative pairs, formed by pairing each image with the text that yields the highest col score from other pairs in the batch. We note the interaction matrices for positive and negative image-text pairs as $\mathbf{X}^+$, $\mathbf{X}^-$, respectively. Finally, we train the MLLM using the InfoCE loss (Oord et al., 2018),

$$\mathcal{L}_{\text{InfoCE}} = -\log\left(\frac{e^{(s(\mathbf{X}^+,\mathbf{M})/\tau)}}{e^{(s(\mathbf{X}^+,\mathbf{M})/\tau)} + e^{(s(\mathbf{X}^-,\mathbf{1})/\tau)}}\right), \tag{3}$$

where $\mathbf{1}$ is a vector of ones. Since there is only one negative pair per sample, the InfoCE loss can be simplified to the Softplus form (Glorot et al., 2011):

$$\mathcal{L} = \text{Softplus}(s(\mathbf{X}^-,\mathbf{M})) - s(\mathbf{X}^+,\mathbf{1}))). \tag{4}$$

**Inference**   During the inference phase, the main goal is to locate the image patches that have the highest similarities with the text embeddings, aligning with the training objective, the text is the concatenation of the question and the ground-truth answer. Unlike tokens that are naturally discrete, the image patches are generated by cutting the continuous image; thus, the information of one object might be unintentionally contained in several adjacent patches, further causing the low similarities of each patch. To alleviate this phenomenon, a novel adjacent patch embedding merging technique is first implemented, in which each image patch embedding $H_i^{\text{Image}}$ will be updated as the average of it and its surrounding embeddings:

$$H_i^{\text{Image'}} = \frac{1}{|\mathcal{N}(i)|} \sum_{j \in \mathcal{N}(i)} H_j^{\text{Image}} \tag{5}$$

where $H_i^{\text{Image'}}$ represents the updated embedding for the $i$th image patch, $\mathcal{N}(i)$ represents the set of indices for the $i$th image patch and its surrounding patches. This adjacent patch embedding merging technique not only completes the semantic information of each patch embedding, but also serves as a smoothing that alleviates the influence of single patches with abnormally high similarities.

After obtaining the updated patch embeddings, a similarity vector $S \in \mathbb{R}^{l_{\text{Image}}}$ can be calculated by first calculating the similarities between each image patch and text token and then averaging over the text embedding space:

$$S_i = \frac{1}{l_{\text{Text}}} \sum_{k=1}^{l_{\text{Text}}} \langle H_i^{\text{Image'}}, H_k^{\text{Text}}\rangle, \quad \forall i \in \{1, \ldots, l_{\text{Image}}\} \tag{6}$$

where $S_i$ represents the $i$th component of the similarity vector, $l_{\text{Text}}$ is the number of text tokens, and $\langle H_i^{\text{Image'}}, H_k^{\text{Text}}\rangle$ represents the dot product between the image and text embeddings.

The similarity vector $S$ represents the similarity of each image patch to the whole text embedding. The image patches with higher similarities contain the information mentioned in the text and thus can be used as the grounding area. Then, the image patches can be found by selecting the top-$k$ highest values. Moreover, to further improve the robustness and alleviate the severe influence of the selection of $k$, a further 2-level selection framework is utilized. In the framework, we first select the image patches with top-5 similarities and then progressively include later top-$k$ patches if it is in the surroundings of the top-5 patches. The detailed ablation studies can be found in the appendix.

## 4 EXPERIMENTAL RESULTS

For the instruction-based method, we utilize LLaVA-v1.5-Vicuna-13B (Liu et al., 2023a) as our base model to be fine-tuned. For the embedding-based method, we utilize PaliGemma-3B (Beyer et al., 2024) as our base model to be fine-tuned. Detailed training configurations can be found in the supplementary material.

## 4.1 MAIN RESULTS

The overall evaluation results across various MLLMs are presented in Table 1 (Evaluation Setting 1), Table 2 (Evaluation Setting 2). The MLLMs we evaluated are listed as follows: LLaVA-v1.6-Vicuna-13B, LLaVA-v1.6-Vicuna-7B (Liu et al., 2023a;b; 2024b), Phi3-V(Team, 2024), DeepSeek-VL-7B-chat (Lu et al., 2024), Idefics2-8B (Laurençon et al., 2023; 2024), Qwen-VL (Bai et al., 2023a), CogVLM2-Llama3-19B (Wang et al., 2023a), InternLM-XComposer2-VL-7B (Dong et al., 2024a), InternLM-XComposer2-4KHD-7B (Dong et al., 2024b), Monkey-Chat (Li et al., 2024b), MiniCPM-Llama3-V 2.5 (Yao et al., 2024), InternVL2 (Chen et al., 2024b), LLaVA-OV (Li et al., 2024a), InternVL2.5 (Chen et al., 2024b), Qwen2.5-VL (Bai et al., 2025), Qwen2-VL (Wang et al., 2024), Gemini-2-flash (DeepMind, 2025b), Gemini-2.5-flash (DeepMind, 2025a), and GPT series.

As shown in Table 1, all the existing models, including open-source models and the powerful GPT4o, do not perform well for the **OCR-free Grounding** setting, in which they are required to generate supportive bounding boxes without any additional bounding box information. Under these circumstances, both our proposed methods outperform GPT4o by a large margin: our embedding-based method reaches the average IoU of 10.0, and our instruction-based method reaches the average IoU of 29.98, compared with 1.51 for GPT4o. Several conclusions can be made from the evaluation of setting 1 results: (1) The capability of existing models to generate grounded bounding boxes from scratch is limited, and thus, the training specifically for this setting is required. (2) Our proposed method is effective, and the quality of our corresponding training dataset is promising as our performance outperforms the powerful GPT4o by a large margin. (3) Instruction tuning on reasonably good LLMs can yield much better performance than utilizing the embedding-based method, while the embedding-based method is much more efficient during inference: Specifically, the instruction method takes an average of 2.37 seconds per document, whereas the Embedding Method takes 0.74 seconds per document, calculated based on inference on 1 Nvidia A100 GPU.

| Testsets | Chart | Doc | Info | Trins | Avg |
|---|---|---|---|---|---|
| Metrics (%) | IoU | IoU | IoU | IoU | Avg |
| LLaVA-v1.6-Vicuna-13B | 0.00 | 0.00 | 0.00 | 0.00 | 0.00 |
| LLaVA-v1.6-Vicuna-7B | 0.00 | 0.00 | 0.00 | 0.00 | 0.00 |
| Phi3-V | 0.00 | 0.00 | 0.00 | 0.00 | 0.00 |
| Qwen2-VL-2B | 0.00 | 0.00 | 0.00 | 0.00 | 0.00 |
| Qwen2-VL-7B | 0.00 | 0.00 | 0.00 | 0.00 | 0.00 |
| Qwen2.5-VL-3B | 0.00 | 0.00 | 0.00 | 0.00 | 0.00 |
| Qwen2.5-VL-7B | 0.00 | 0.00 | 0.00 | 0.00 | 0.00 |
| LLaVA-OV-1B | 0.00 | 0.00 | 0.00 | 0.00 | 0.00 |
| LLaVA-OV-7B | 0.00 | 0.00 | 0.00 | 0.00 | 0.00 |
| InternVL2-1B | 0.00 | 0.00 | 0.00 | 0.00 | 0.00 |
| InternVL2-8B | 0.00 | 0.00 | 0.00 | 0.00 | 0.00 |
| InternVL2.5-1B | 0.00 | 0.00 | 0.00 | 0.00 | 0.00 |
| InternVL2.5-8B | 0.00 | 0.00 | 0.00 | 0.00 | 0.00 |
| DeepSeek-VL-7B-chat | 0.07 | 0.00 | 0.02 | 0.00 | 0.02 |
| Idefics2-8B | 0.21 | 0.01 | 0.00 | 0.00 | 0.06 |
| Qwen-VL-7B | 0.43 | 0.06 | 0.18 | 0.23 | 0.22 |
| CogVLM2-Llama3-19B | 0.19 | 0.01 | 0.16 | 0.66 | 0.25 |
| InternLM-XComposer2-VL-7B | 0.15 | 0.20 | 0.13 | 0.57 | 0.26 |
| Monkey-Chat | 0.77 | 0.19 | 0.15 | 0.45 | 0.39 |
| InternLM-XComposer2-4KHD-7B | 1.04 | 0.10 | 0.90 | 0.14 | 0.55 |
| MiniCPM-Llama3-V 2.5 | 0.44 | 1.40 | 0.65 | 4.96 | 1.86 |
| Gemini2.0-Flash | 0.00 | 0.00 | 0.00 | 0.00 | 0.00 |
| Gemini2.5-Flash | 0.00 | 0.00 | 0.00 | 0.00 | 0.00 |
| GPT-4o | 3.90 | 1.79 | 1.60 | 13.73 | 5.26 |
| **Ours (Embedding)** | 10.51 | 15.02 | 7.85 | 13.88 | 11.82 |
| **Ours (Instruction)** | **27.91** | **23.96** | **8.61** | **59.44** | **29.98** |

Table 1: **OCR-free Grounding.** Chart, Doc, Info, and Trins represent evaluation results on ChatQA, DocVQA, InfographicsVQA, and TRINS datasets, respectively. IoU represents the pixel-level IoU scores. Avg represents the average IoU score on the 4 datasets, and the ordering of each model is decided by this average score.

As shown in Table 2, for the **OCR-based Grounding** setting, most of the models can reach a much better performance with the OCR information provided. However, the performances of open-source models still have a large gap with GPT4o. Both of our methods achieve better performances compared with the existing open-source models while failing to beat GPT4o, which is due to its inherently powerful instruction-following capability for customized instructions.

## 4.2 FURTHER FINDINGS

> **Finding 1.** All existing MLLMs are not good at generating grounded bounding boxes from scratch.

As shown in Table 1, even the most powerful GPT4o can only gain an average IoU score of 5.28%, consisting of 13.73% on TRINS and approximately under 4.0% on other datasets. The relatively higher performance on TRINS is due to its special characteristic that the images in TRINS contain the least number of OCR objects while occupying the most area, making it easier to generate intersected bounding boxes. However, when it comes to common information-intensive documents, the performances drop dramatically as the ground truth bounding boxes become much smaller.

| Testsets | ChartQA | | | | DocVQA | | | | InfographicsVQA | | | | TRINS | | | | Avg |
|---|---|---|---|---|---|---|---|---|---|---|---|---|---|---|---|---|---|
| Metrics (%) | IoU | P | R | F1 | IoU | P | R | F1 | IoU | P | R | F1 | IoU | P | R | F1 | Avg |
| LLaVA-OV-7B | 0.52 | 0.63 | 0.77 | 0.68 | 0.25 | 0.33 | 0.50 | 0.40 | 0.10 | 0.17 | 0.17 | 0.17 | 1.89 | 2.33 | 2.29 | 2.12 | 0.84 |
| LLaVA-v1.6-Vicuna-13B | 0.00 | 0.00 | 0.00 | 0.00 | 0.00 | 0.00 | 0.00 | 0.00 | 0.00 | 0.00 | 0.00 | 0.00 | 0.00 | 0.00 | 0.00 | 0.00 | 0.00 |
| LLaVA-v1.6-Vicuna-7B | 0.00 | 0.00 | 0.00 | 0.00 | 0.00 | 0.00 | 0.00 | 0.00 | 0.00 | 0.00 | 0.00 | 0.00 | 0.00 | 0.00 | 0.00 | 0.00 | 0.00 |
| Idefics2-8b | 0.00 | 0.00 | 0.00 | 0.00 | 0.00 | 0.00 | 0.00 | 0.00 | 0.00 | 0.00 | 0.00 | 0.00 | 1.17 | 1.17 | 3.50 | 1.69 | 0.47 |
| DeepSeek-VL-7B-chat | 0.84 | 1.35 | 2.25 | 1.23 | 0.19 | 0.19 | 1.50 | 0.34 | 0.00 | 0.00 | 0.00 | 0.00 | 1.60 | 1.60 | 2.00 | 1.67 | 0.92 |
| InternLM-XComposer2-4KHD-7B | 1.00 | 2.00 | 1.00 | 1.25 | 0.25 | 0.50 | 0.25 | 0.33 | 0.00 | 0.00 | 0.00 | 0.00 | 3.67 | 6.80 | 3.79 | 4.65 | 1.59 |
| Monkey-Chat | 3.58 | 5.38 | 9.91 | 5.05 | 0.94 | 1.19 | 1.75 | 1.15 | 0.34 | 0.30 | 2.00 | 0.51 | 0.00 | 0.00 | 0.00 | 0.00 | 2.01 |
| Phi3-V | 2.54 | 3.14 | 5.46 | 3.25 | 0.97 | 1.22 | 2.00 | 1.21 | 0.40 | 0.35 | 2.08 | 0.59 | 3.81 | 4.31 | 5.00 | 4.28 | 2.54 |
| CogVLM2-Llama3-19B | 1.65 | 2.30 | 3.68 | 2.02 | 1.56 | 1.87 | 3.58 | 1.89 | 0.42 | 0.84 | 1.75 | 0.71 | 6.76 | 7.93 | 8.33 | 7.53 | 3.30 |
| InternVL2-8B | 0.07 | 0.50 | 0.07 | 0.13 | 3.92 | 4.62 | 6.33 | 4.50 | 1.11 | 1.63 | 1.42 | 1.36 | 5.60 | 6.39 | 6.39 | 6.28 | 3.10 |
| Qwen-VL-7B | 4.36 | 4.33 | 28.00 | 7.27 | 1.07 | 1.06 | 8.25 | 1.83 | 0.70 | 0.71 | 4.50 | 1.20 | 1.51 | 1.83 | 3.32 | 2.08 | 4.50 |
| Qwen2-VL-2B | 5.21 | 6.65 | 11.56 | 7.86 | 3.48 | 3.67 | 10.25 | 5.25 | 0.49 | 0.75 | 1.00 | 0.84 | 7.23 | 8.13 | 11.17 | 8.79 | 4.52 |
| LLaVA-OV-1B | 3.57 | 4.45 | 7.90 | 5.36 | 4.24 | 4.13 | 9.88 | 5.59 | 0.51 | 0.75 | 1.08 | 0.86 | 7.05 | 8.21 | 10.79 | 8.65 | 4.86 |
| Qwen2.5-VL-3B | 3.77 | 5.74 | 5.37 | 5.02 | 2.46 | 3.00 | 3.58 | 3.02 | 0.10 | 0.17 | 0.17 | 0.17 | 10.19 | 12.75 | 14.59 | 12.34 | 5.05 |
| InternVL2.5-VL-1B | 3.01 | 2.91 | 16.19 | 4.70 | 0.56 | 0.55 | 5.25 | 0.99 | 0.03 | 0.03 | 0.50 | 0.06 | 14.47 | 16.12 | 33.54 | 19.20 | 8.16 |
| InternLM-XComposer2-VL-7B | 8.10 | 15.31 | 9.03 | 10.49 | 7.36 | 11.28 | 8.08 | 8.73 | 2.32 | 6.39 | 2.85 | 3.39 | 15.82 | 18.12 | 17.32 | 16.98 | 10.10 |
| Qwen2-VL-7B | 14.49 | 20.13 | 20.25 | 18.73 | 14.43 | 16.29 | 22.50 | 17.37 | 2.22 | 2.91 | 4.13 | 3.33 | 10.22 | 11.00 | 13.60 | 11.43 | 11.34 |
| InternVL2-1B | 3.43 | 3.28 | 19.10 | 5.35 | 0.10 | 0.10 | 0.75 | 0.17 | 0.35 | 0.35 | 2.71 | 0.62 | 23.91 | 32.30 | 48.38 | 31.10 | 15.56 |
| MiniCPM-Llama3-V 2.5 | 7.40 | 12.50 | 8.40 | 9.24 | 15.78 | 19.22 | 18.42 | 17.48 | 5.71 | 9.87 | 7.34 | 7.14 | 54.06 | 62.06 | 57.89 | 57.95 | 23.15 |
| InternVL2.5-8B | 9.99 | 12.54 | 14.24 | 11.51 | 5.71 | 6.50 | 7.58 | 6.25 | 5.11 | 8.06 | 8.46 | 6.70 | 59.51 | 63.98 | 70.73 | 64.05 | 28.82 |
| Qwen2.5-VL-7B | 16.84 | 26.04 | 18.61 | 20.22 | 24.12 | 27.33 | 25.75 | 25.80 | 5.81 | 11.42 | 6.40 | 7.69 | 55.12 | 66.21 | 66.45 | 62.80 | 30.19 |
| Gemini2.0-Flash | 45.76 | 49.38 | 50.01 | 48.25 | 57.36 | 64.02 | 61.38 | 60.88 | 40.18 | 47.35 | 47.80 | 45.04 | 32.04 | 35.62 | 33.75 | 33.67 | 44.46 |
| **Ours (Embedding)** | 39.97 | 57.26 | 52.89 | 49.98 | 37.82 | 40.09 | 72.96 | 48.42 | 25.58 | 28.68 | 49.14 | 32.93 | 70.01 | 86.38 | 75.94 | 77.32 | 52.84 |
| **Ours (Instruction)** | 70.38 | 81.58 | 73.78 | 75.78 | 73.52 | 81.67 | 75.13 | 77.02 | 39.23 | 47.29 | 42.86 | 43.90 | 85.48 | 92.17 | 79.94 | 83.62 | 69.77 |
| Gemini2.5-Flash | 74.65 | 81.31 | 81.06 | 78.92 | 73.67 | 81.86 | 78.00 | 77.71 | 73.11 | 79.58 | 73.12 | 72.78 | 78.43 | 76.85 | 75.80 | | 75.23 |
| GPT-4o | 83.80 | 88.80 | 89.24 | 87.47 | 82.14 | 87.14 | 89.50 | 86.16 | 68.19 | 79.57 | 78.81 | 75.82 | 89.08 | 96.06 | 91.53 | 92.16 | 85.34 |
| GPT-3.5-turbo (Without Image) | 8.81 | 14.17 | 9.48 | 10.64 | 32.05 | 40.92 | 32.50 | 35.00 | 12.66 | 20.45 | 14.39 | 15.75 | 15.81 | 18.75 | 15.80 | 16.62 | 19.58 |
| GPT-4 (Without Image) | 51.58 | 57.29 | 53.75 | 54.34 | 52.18 | 62.79 | 53.61 | 56.28 | 47.31 | 57.51 | 54.34 | 53.51 | 69.83 | 76.16 | 71.05 | 72.39 | 58.59 |
| GPT-4o (Without Image) | 59.50 | 67.13 | 62.84 | 63.29 | 77.83 | 83.41 | 80.71 | 80.80 | 63.34 | 72.64 | 69.03 | 68.88 | 71.05 | 80.01 | 72.54 | 74.38 | 71.69 |

Table 2: **OCR-based Grounding.** IoU, P, R, F1 represent bounding-box-level IoU score, precision, recall and F1 score. Avg represents the average score on all datasets and evaluation metrics, and the ordering is decided by this score.

Compared with GPT4o, the IoU values from other open-source models are mostly under $1.0\%$, which is negligible. GPT4o is able to follow the instructions and generate bounding boxes, though its relatively weak spatial understanding makes the generation not precise enough. However, most of the other open-source models are not able to either understand our instructions or generate reasonable bounding boxes, especially for those MLLMs that get near-zero IoU values. Even if we provide further instructions requiring them to generate bounding boxes that can support their answer, they are not able to understand the instructions or follow them.

These results reveal a critical issue: current MLLMs have a relatively weak spatial understanding and are not capable of generating grounded boxes that support their answer from scratch, which makes it a potential future direction.

> **Finding 2.** Most existing open-source MLLMs are not able to follow customized complex instructions.

Table 2 represents the evaluation setting where the OCR information, including bounding boxes and texts, is wrapped into the input to MLLMs, thus making the whole process a much simpler bounding box selection process. The performance of GPT4o reaches an astonishingly high value of $85.34\%$ on average compared with the $5.26\%$ on evaluation setting 1, indicating that generating grounded bounding boxes from scratch is hard for GPT4o. In this setting, even the text-only models can achieve a reasonably high performance due to the detailed information provided by OCR models, while the performances of most of the existing open-source MLLMs are still kept low, making it impossible for practical usage.

By careful inspection, we observe that these low performances on existing open-source MLLMs are mainly caused by their inability to follow the given instructions: most existing open-source MLLMs will directly generate corresponding answers to the question and ignore the instruction of selecting supporting grounded bounding boxes, potentially due to the overfitting on the format of the training data. To further quantitatively analyze this issue, we introduce another value, **the instruction-following rate**, defined by the proportion of testing samples for which the MLLMs are able to generate at least one bounding box required in the additional instruction. This value does not measure the correctness of MLLM-generated bounding boxes but only their existence. It directly represents MLLMs' instruction-following ability for this task, i.e., generate at least one bounding box regardless of the correctness.

The results of the instruction-following rates of different models are presented in the supplementary material. GPT4o reaches an average instruction-following rate of $98\%$. Even if for the OCR-free Grounding setting, in which it can only reach an average IoU score of $5.26\%$, it still achieves the instruction-following rate of $96\%$. The comparison between the instruction-following rate and IoU

score reveals that (i) GPT4o has a strong capability in following customized complex instructions; (ii) The low IoU is caused by GPT4o's inability of spatial relationships. On the contrary, the instruction-following rates on existing open-source MLLMs are mostly less than 30.0% or even 10.0%, which means that under most circumstances, they do not understand our instruction and do not generate any bounding boxes, representing a "not-even-wrong" situation. Since the average instruction-following rates are much higher than the grounding metrics, the gaps between them indicate the MLLMs' inability to ground.

Our settings reflect both MLLMs' instruction-following and grounding ability. Most of the existing open-source models are not able to follow the relatively complex instructions in this task, and even if they correctly follow the instructions, most of them are not able to find the correct grounded bounding boxes that support their answers. Comparing these results with their outstanding performances on standard QA tasks, we reveal this critical issue that most existing open-source MLLMs are potentially overfitted to the standardized tasks, and they still lack instruction-following abilities.

## 5 FURTHER DISCUSSIONS

### 5.1 PRACTICAL USE CASE

This text-rich document's grounding capability helps users understand the source of the information provided in the answers, **building users' trust in the system**. When users ask questions to AI assistants about documents, they need to verify the answer. Without grounding, they may look through the whole document, which damages the user experience. However, if grounded boxes are given, users can verify the answer with only a glimpse, largely **accelerating users' reliance on AI assistants for their daily work**. Besides, text grounding has been deployed in real-world products, while visual text grounding remains challenging. Thus, this benchmark is practically useful.

### 5.2 DIFFERENCES BETWEEN COMMON TEXT LOCALIZATION TASK

Our task focuses on text grounding in document images, which is **different from text localization and orthogonal to all existing tasks**: Text localization requires the model to provide the location of specific visual text or extract it from the page. For example, the text localization used in KOSMOS-2.5 Lv et al. (2023) contains Document-Level Text Recognition and Image-to-Markdown Generation, which are totally different from ours. On the contrary, our visual text grounding aims to provide the answer with its sources from the image. **The key difference lies in that our prediction of text position is coupled with the understanding of image content.**

## 6 CONCLUSION

In this paper, we introduce TRIG, a novel task designed to evaluate and enhance the visual text grounding capabilities of MLLMs in text-rich document images. We first built TRIG-Bench, a comprehensive benchmark and a large-scale instruction tuning dataset. Our evaluation reveals significant limitations in the visual text grounding capabilities of existing MLLMs, particularly in their ability to handle complex document layouts and textual content. Hence, we propose two simple and effective instruction-tuning-based and embedding-based methods that demonstrate substantial improvements over existing models.

## REPRODUCIBILITY STATEMENT

All the prompts we used for generation and evaluation in this paper, along with the benchmark data statistics, have been provided in the Appendix C and H. Moreover, all the code and data are available at `https://anonymous.4open.science/r/TRIG-Bench-5BD8/`.

## THE USE OF LARGE LANGUAGE MODELS

In preparing this manuscript, we used LLM exclusively as a writing assistant. The model was employed to suggest improvements to sentence structure, readability, and stylistic consistency. After

receiving suggestions from LLM, the authors carefully reviewed, revised, and further modified the text. LLM did not contribute to research ideation, methodology design, data analysis, or the generation of any scientific content. All substantive intellectual contributions, interpretations, and conclusions remain entirely the responsibility of the authors.

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

TABLE OF CONTENTS FOR APPENDIX

## A    RELATED WORKS

**MLLM for Document Understanding**    In the area of document understanding, driven by the remarkable capabilities of multimodal large language models in visual-language tasks (Tiong et al., 2022; Driess et al., 2023), generative models (Zhang et al., 2023b; Ye et al., 2023a) are gradually replacing traditional supervised techniques (Xu et al., 2020b; Wang et al., 2021; Xu et al., 2020a; Yu et al., 2023). Among them, LLaVAR (Zhang et al., 2023b) utilizes GPT-4 (OpenAI, 2023) with OCR models to generate diverse instruction-following data for text-rich images for further training on LLaVA (Liu et al., 2023b). mPLUG-DocOwl (Ye et al., 2023a), as an evolved version of mPLUG-Owl (Ye et al., 2023c), is designed for the scenario with dense textual content, which achieves outstanding performance in various downstream tasks without OCR. UniDoc (Feng et al., 2023) mainly addresses potential mismatches between pre-training and fine-tuning data distributions by utilizing PowerPoint presentations to create OCR-related instruction-following data. Additionally, significant research efforts such as UReader (Ye et al., 2023b) and KOSMOS-2.5 (Lv et al., 2023) also make diverse contributions to the field.

**MLLM for Visual Grounding**    Grounding (Harnad, 1990) is essential for effective human communication with machines. Currently, grounding datasets have been utilized for vision-language pre-training and improvement for both object-level recognition (Li et al., 2022) and language acquisition (Ma et al., 2023). Recent studies propose to integrate text and grounding regions into token sequences (Yang et al., 2022; Lu et al., 2022; Wang et al., 2022) within language modeling frameworks. Building on such approaches, researchers have developed a series of grounded MLLMs, including GPT4ROI (Zhang et al., 2023a), Kosmos-2 (Peng et al., 2023), Shikra (Chen et al., 2023b), PVIT (Chen et al., 2023a), BuboGPT (Zhao et al., 2023), Qwen-VL (Bai et al., 2023b), and Ferret (You et al., 2023). Despite their impressive performance, these models mainly focus on the grounding of natural image objects, and the grounding of text-rich document images is still under-explored. For the two most related works that also mentioned text grounding on document images, $P^2G$ (Chen et al., 2024a) includes the OCR model into the QA pipeline as an information amplifier; TG-Doc (Wang et al., 2023b) utilizes grounding capability to improve models' QA capabilities. Both of these works treat the text grounding process as an intermediate step and do not conduct any evaluation of the grounding capability. On the contrary, we entirely focus on the grounding capability and first propose the benchmark for it. Another related work is TextMonkey (Liu et al., 2024d), which describes a task that is similar to ours, but there are no examples, illustrations, or systematic evaluations of this ability included. This situation further showcases the value of our benchmark.

**MLLM Limitations in Grounding**    A series of recent models have advanced visual grounding and text-centric document understanding. SPTS [1] focuses on end-to-end scene text spotting with extremely low-cost single-point annotations, treating text detection and recognition as a sequence prediction problem in natural images, but it does not address document reasoning or answer-support grounding in text-rich documents. The DocOwl series [2] proposes unified structure learning for OCR-free document understanding and introduces multi-grained text localization tasks over documents, tables, charts, webpages, and natural images; however, these localization tasks are defined at the level of text spans (word/phrase/line/block) and are used as pretraining objectives for structure-aware parsing and recognition, rather than evaluating which regions support a downstream QA answer. KOSMOS-2.5 [3] pre-trains a "multimodal literate" model on large-scale document-level text recognition and image-to-Markdown generation, producing spatially-aware text blocks (text + bounding-box coordinates) and structured markdown outputs across diverse document types, but its grounding remains tied to transcription and layout rather than question-conditioned evidence localization. TokenVL builds on a token-level text-image foundation model (TokenFD) and uses token masks to train fine-grained image-as-text alignment and VQA-based text parsing; its grounding signals come from token-level masks (or text spans) and are evaluated via OCR/VQA performance, not via explicit answer-support region accuracy. Marten similarly introduces a VQAMask pretraining paradigm that combines VQA-based text parsing with mask generation to improve spatial awareness in document-level MLLMs, but the mask supervision is still derived from generic text masks, and its downstream evaluation focuses on text-centric VQA and OCRBench, rather than on identifying the specific regions that justify answers to document QA. Although these advances push visual grounding in text-rich scenarios forward, a key research gap remains largely unaddressed: existing methods do not tackle question-conditioned evidence localization. They focus on text spotting, structure parsing,

spatially-aware text generation, or pretraining-based alignment, but none require models to identify which specific region in a document provides the supporting evidence for a given question. This missing capability leaves open an important direction for understanding and evaluating grounding in text-rich document images.

## B  EMBEDDING-BASED METHOD ILLUSTRATION

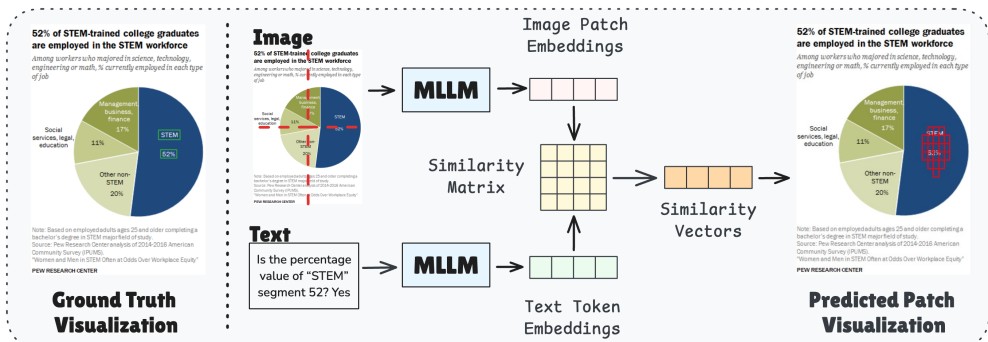

Figure 3: The pipeline of our embedding-based method. The example is from ChartQA. **Question:** Is the percentage value of the "STEM" segment 52? **Answer:** Yes. The visualization of ground-truth bounding boxes is presented on the left in green, and the visualization of the generated grounding area (patches) by our embedding-based method is presented on the right in red. The input image will be processed into $32 \times 32$ patches before sending it into the MLLM and obtaining the image patch embeddings. After obtaining both the image patch embeddings and text token embeddings, a similarity vector with a length of the number of image patches is generated. The higher scores represent the alignment between image and text, whose position will be selected as the grounding patches. For simplicity, the embedding, merge, and 2-level selection mechanisms are not presented in this figure.

Motivated by the finding that the embeddings representing similar contents have relatively higher similarity scores (Faysse et al., 2024), we propose this method that does not count on the instruction-following capability of MLLMs. An illustrative figure for our embedding-based method pipeline is shown in Figure 3. Different from the instruction-based method requiring MLLMs to generate tokens iteratively, which is extremely time-consuming for a large number of forward processes, the embedding-based method only needs two times of forward processes, which largely increases the efficiency of the grounding process.

## C  DATA INFORMATION

### C.1  DATA SOURCES

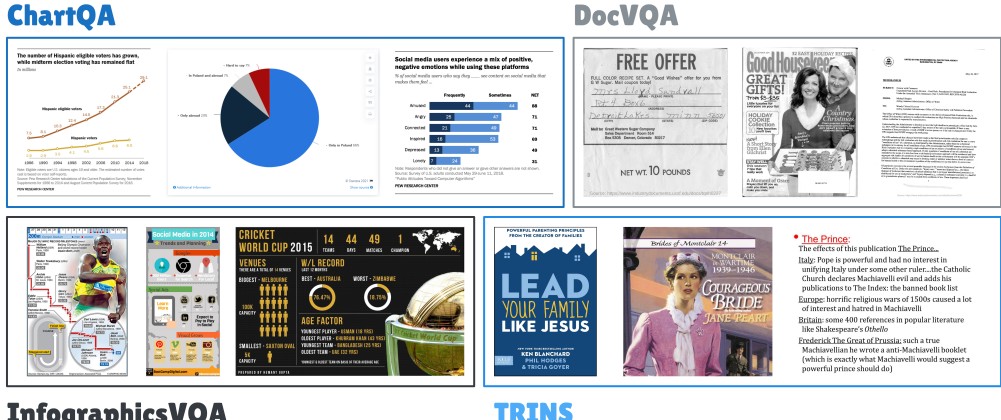

Figure 4: Text-rich document image examples from different source datasets.

|  | Chart | Doc | Info | Trins | Total |
|---|---|---|---|---|---|
| Total Question # | 200 | 200 | 200 | 200 | 800 |
| Total Image # | 171 | 190 | 199 | 199 | 759 |
| Avg Question Len | 11.04 | 9.66 | 12.42 | 11.69 | 11.20 |
| Avg Answer Len | 1.39 | 2.94 | 1.93 | 19.48 | 6.44 |
| Avg OCR Text Len | 2.7 | 4.73 | 3.02 | 3.00 | 3.36 |
| Avg OCR Box # | 26.15 | 53.30 | 102.26 | 8.94 | 47.66 |
| Avg GT Box # | 2.67 | 1.73 | 2.78 | 2.45 | 2.41 |
| Avg GT Box Ratio | 11.83% | 4.88% | 3.72% | 37.74% | 14.54% |
| Avg OCR Area (%) | 12.56 | 19.75 | 18.29 | 15.82 | 16.61 |
| Avg GT Area (%) | 0.80 | 0.79 | 0.53 | 6.09 | 2.05 |
| Avg GT Area Ratio | 8.59% | 5.10% | 2.89% | 42.69% | 14.82% |

Table 3: **Benchmark Data Statistics.** Total Question # represents the unique question number from each dataset. Total Image # represents the Unique image number. Other statistics are averaged on each dataset.

To ensure the diversity of our benchmark data, four existing document datasets are chosen as our data sources, covering a wide range of document image types, including **DocVQA** (Mathew et al., 2021), **ChartQA** (Masry et al., 2022), **InfographicVQA** (Mathew et al., 2022), and **TRINS** (Zhang et al., 2024a) as shown in Figure 4.

**DocVQA** (Mathew et al., 2021) is designed to test the capabilities of VQA systems in the domain of document understanding. The dataset comprises a diverse collection of document images, including invoices, forms, receipts, reports, letters, historical documents, etc. The questions are crafted to test various aspects of document understanding, such as information retrieval, layout interpretation, and relational reasoning. Our training data source is the whole DocVQA training subset. Due to the unavailability of the test subset, our benchmark data is selected from the validation subset.

**ChartQA** (Masry et al., 2022) is designed to assess the proficiency of VQA systems in interpreting data visualizations. It features a wide array of chart types, including bar charts, pie charts, line graphs, scatter plots, histograms, and box plots. It is particularly focused on the task of understanding and reasoning about quantitative information presented in visual formats. The whole ChartQA training subset is used as our training data source and our benchmark data is selected from the test subset.

**InfographicVQA** (Mathew et al., 2022) is a dataset that explores the realm of visually rich and diverse infographics, such as posters, flyers, advertisements, educational infographics, and infographic-style presentations. The diverse nature of the infographics in this dataset, coupled with the complexity of the questions, makes it a crucial resource for developing VQA systems capable of understanding and interacting with complex visual information. The whole InfographicVQA training subset is used as our training data source. Due to the unavailability of the test subset, our benchmark data is selected from the validation subset.

**TRINS** (Zhang et al., 2024a) is a novel dataset designed to enhance the ability of multimodal language models to read and understand text-rich images. This dataset addresses the limitation of existing visually-tuned models, which often struggle with textual comprehension within images. It includes text-rich images, such as posters and book covers, providing a significant challenge due to the high density and complexity of text present in each image. The dataset supports various tasks, including visual captioning, visual question answering, and text-rich image generation. We mainly utilize its VQA subset. Our benchmark data is randomly sampled from the whole dataset, and the remaining data is used as our training data source.

## C.2 Data Statistics

The benchmark data statistics are presented in Table 3. "Avg Question Len", "Avg Answer Len", and "Avg OCR Text Len" calculate the average question, answer, and OCR text word counts. "Avg OCR Box #", "Avg GT Box", and "Avg GT Box Ratio" calculate the number of bonding boxes provided by OCR models, selected as the ground truth, and their ratio. "Avg OCR Area (%)", "Avg GT Area (%)", and "Avg GT Area Ratio" calculate the area of bonding boxes provided by OCR models, selected as the ground truth, and their ratio. From these statistics, the characteristics of each

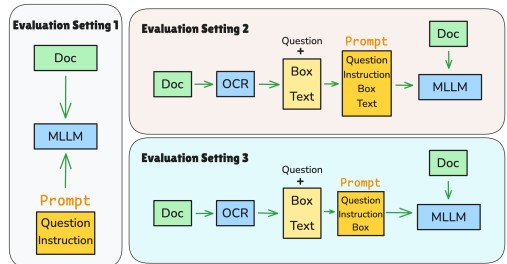

Figure 5: **Illustrations on Evaluation Settings.** In evaluation setting 1, no OCR model is used, representing the hardest scenario. While in settings 2 & 3, an additional OCR model is utilized to facilitate LLM on grounding information generation. The "Instruction" in the prompt describes the requirement of generating grounded bounding boxes and defines the desired format.

dataset can be illustrated, which showcases the variety of our data components and further provides a clear understanding of evaluation results.

## D  TRAINING CONFIGURATIONS

Our training data contains $22,363$ samples from CharQA, $29,094$ samples from DocVQA, $19,840$ samples from InfographisVQA, and $24,885$ samples from TRINS. All experiments are performed on Nvidia A100 GPUs. We utilize $8$ GPUs for training and $1$ for inference.

For the **instruction-based method**, we utilize LLaVA-v1.5-Vicuna-13B (Liu et al., 2023a) as our base model to be finetuned, based on the codebase from Vicuna (Chiang et al., 2023) and LLaVA (Liu et al., 2023b). We train our model with a batch size of $128$ and a max token length of $8192$. The maximum learning rate is set to $2 \times 10^{-5}$ with a warmup rate of $0.03$ for 3 epochs.

For the **embedding-based method**, we utilize PaliGemma-3B (Beyer et al., 2024) as our base model to be finetuned, based on the codebase from ColPali (Faysse et al., 2024). We train our model for $4$ epochs with a batch size of $32$ and a learning rate of $5e-5$, using the AdamW optimizer. LoRA adapters with a rank of $32$ are added to all MLP layers in the LLM.

## E  DETAILED EVALUATION SETTINGS AND METRICS

To simulate various real-world scenarios and test models' capability for grounding at different levels, three different evaluation settings are provided that are compatible with this benchmark.

### E.1  EVALUATION SETTING 1 (OCR-FREE GROUNDING)

In this setting, as shown in Figure 5 (a), only the document image and corresponding question are provided for the MLLMs, together with the specific instruction that describes the task requirement and defines the desired format. It represents the hardest level of MLLMs' grounding capabilities as it requires MLLMs to answer the question and simultaneously generate the corresponding grounded bounding boxes from scratch. In addition to the measurement of MLLMs' grounding capabilities, it also measures their instruction-following abilities. As a complex, customized task, the instructions for generating grounded bounding boxes have never been seen by MLLMs during training, and it requires deep reasoning capability to correctly follow.

**Pixel-level IoU** (Intersection over Union) is utilized as the evaluation metric. It calculates the overlap between pixels in the predicted bounding boxes and ground truth bounding boxes, normalized by the total number of unique pixels. It is commonly used in image segmentation tasks to assess the accuracy of pixel-wise predictions:

$$\text{IoU}_{\text{pixel}} = \frac{1}{N} \sum_{i=1}^{N} \frac{|\text{Pred}_i \cap \text{GroundTruth}_i|}{|\text{Pred}_i \cup \text{GroundTruth}_i|} \tag{7}$$

where $N$ represents the total number of testing samples, $\text{Pred}_i$ represents the pixels within the predicted grounded bounding boxes of $i$th testing sample and $\text{GroundTruth}_i$ represents the pixels within the groundtruth bounding boxes.

### E.2 EVALUATION SETTING 2 (OCR-BASED GROUNDING)

Although the previous setting aligns with our aim the best, it is too hard for all the existing models, from proprietary models to open-source models. Thus, further evaluation settings are proposed. In this setting, as shown in Figure 5 (b), an additional OCR model is utilized to facilitate the generation of grounded bounding boxes. Specifically, all the bounding box coordinates and corresponding text content obtained from the OCR model will be wrapped into the prompt for MLLMs as additional information. Under these circumstances, the bounding box generation task is converted into an easier bounding box selection/retrieval task, in which MLLMs do not need to generate coordinates from scratch but only need to select the proper ones given in the prompt. It is worth noting that this evaluation setting still measures MLLMs' grounding capability as they have to align the given OCR information to the image information. In this setting, instance-level IoU score, precision, recall, and F1 score are utilized as the evaluation metrics.

**Instance-level IoU** measures the overlap between the retrieved grounded bounding boxes and the ground truth bounding boxes, normalized by the total number of unique elements. It evaluates how well the retrieved bounding boxes match the ground truth ones:

$$\text{IoU}_{\text{inst}} = \frac{1}{N} \sum_{i=1}^{N} \frac{|\text{Pred}_i \cap \text{GroundTruth}_i|}{|\text{Pred}_i \cup \text{GroundTruth}_i|} \tag{8}$$

where $N$ represents the number of testing samples, $\text{Pred}_i$ represents the bounding boxes selected from the prompt of $i$th testing sample and $\text{GroundTruth}_i$ represents the groundtruth bounding boxes.

**Precision** measures the proportion of correctly retrieved elements out of all elements retrieved by the model. It reflects the accuracy of the model's positive predictions:

$$\text{Precision} = \frac{1}{N} \sum_{i=1}^{N} \frac{|\text{Pred}_i \cap \text{GroundTruth}_i|}{|\text{Pred}_i|} \tag{9}$$

**Recall** measures the proportion of correctly retrieved elements out of all actual ground truth elements. It indicates the model's ability to capture all relevant elements in the ground truth.

$$\text{Recall} = \frac{1}{N} \sum_{i=1}^{N} \frac{|\text{Pred}_i \cap \text{GroundTruth}_i|}{|\text{GroundTruth}_i|} \tag{10}$$

**F1 Score** is the harmonic mean of precision and recall, providing a balanced measure that accounts for both false positives and false negatives.

$$\text{F1} = \frac{1}{N} \sum_{i=1}^{N} \frac{2 \times \text{Precision}_i \times \text{Recall}_i}{\text{Precision}_i + \text{Recall}_i} \tag{11}$$

where $\text{Precision}_i$ and $\text{Recall}_i$ represent the precision and recall of $i$th testing sample.

### E.3 EVALUATION SETTING 3

Although the above two settings represent the most common scenario of grounded bounding box generation for document-level VQA tasks, they both have their own flaws: Setting 1 is too difficult for existing MLLMs if not specifically trained on in-domain data, and even GPT4o can only have a performance under 10%, thus not suitable for the evaluation of most existing MLLMs. Setting 2, on the contrary, provides a potential shortcut that models might directly select the bounding boxes according to the text content.

Thus, Setting 3, as shown in Figure 5 (c), is proposed as the combination of previous settings, where the OCR-generated bounding boxes will be provided without the specific text content. Under these

| Metrics | ChartQA | | | | DocVQA | | | | InfographicsVQA | | | | TRINS | | | | Avg |
|---|---|---|---|---|---|---|---|---|---|---|---|---|---|---|---|---|---|
| Testsets | IoU | P | R | F1 | IoU | P | R | F1 | IoU | P | R | F1 | IoU | P | R | F1 | Avg |
| Idefics2-8b | 0.00 | 0.00 | 0.00 | 0.00 | 0.00 | 0.00 | 0.00 | 0.00 | 0.00 | 0.00 | 0.00 | 0.00 | 0.00 | 0.00 | 0.00 | 0.00 | 0.00 |
| DeepSeek-VL-7B-chat | 0.38 | 0.38 | 3.00 | 0.67 | 0.08 | 0.08 | 1.00 | 0.16 | 0.03 | 0.03 | 0.13 | 0.05 | 0.00 | 0.00 | 0.00 | 0.00 | 0.37 |
| InternLM-XComposer2-4KHD-7B | 0.23 | 0.26 | 0.83 | 0.36 | 0.00 | 0.00 | 0.00 | 0.00 | 0.22 | 0.26 | 2.42 | 0.38 | 2.13 | 5.13 | 2.43 | 2.97 | 1.10 |
| CogVLM2-Llama3-19B | 0.36 | 0.37 | 3.17 | 0.64 | 0.31 | 0.31 | 3.50 | 0.55 | 0.42 | 0.46 | 2.50 | 0.73 | 3.54 | 4.60 | 4.67 | 4.67 | 2.09 |
| Phi-3-V | 1.12 | 1.36 | 5.92 | 1.81 | 0.69 | 0.69 | 1.50 | 0.81 | 0.32 | 0.45 | 2.29 | 0.56 | 6.05 | 6.38 | 11.50 | 7.17 | 3.04 |
| LLaVA-v1.6-Vicuna-7B | 0.78 | 0.93 | 5.56 | 1.35 | 0.77 | 0.75 | 9.33 | 1.35 | 0.50 | 0.52 | 3.20 | 0.88 | 3.55 | 3.56 | 11.83 | 5.08 | 3.12 |
| InternLM-XComposer2-VL-7B | 2.03 | 3.51 | 5.29 | 3.00 | 2.35 | 3.00 | 2.50 | 2.67 | 0.48 | 1.14 | 2.83 | 0.77 | 8.70 | 12.93 | 10.10 | 10.26 | 4.47 |
| Qwen-VL | 4.25 | 4.21 | 35.82 | 7.35 | 1.14 | 1.13 | 11.75 | 2.02 | 0.80 | 0.82 | 7.60 | 1.46 | 0.89 | 0.89 | 2.50 | 1.20 | 5.24 |
| Monkey-chat | 6.15 | 6.46 | 40.06 | 10.30 | 1.51 | 1.50 | 18.58 | 2.72 | 1.25 | 1.26 | 11.92 | 2.25 | 0.70 | 0.70 | 1.00 | 0.79 | 6.70 |
| MiniCPM-Llama3-V 2.5 | 2.40 | 3.61 | 3.92 | 3.04 | 4.86 | 6.02 | 5.83 | 5.21 | 0.48 | 0.81 | 1.63 | 0.70 | 23.82 | 41.41 | 27.08 | 29.46 | 10.02 |
| LLaVA-v1.6-Vicuna-13B | 6.21 | 6.49 | 35.55 | 10.01 | 1.72 | 1.72 | 6.00 | 2.20 | 0.76 | 0.74 | 5.11 | 1.26 | 15.58 | 16.54 | 39.08 | 20.85 | 10.61 |
| Qwen2-2B | 5.35 | 7.37 | 11.34 | 7.69 | 2.18 | 2.23 | 8.75 | 3.48 | 0.57 | 0.75 | 1.21 | 0.88 | 4.09 | 4.90 | 6.75 | 5.05 | 4.21 |
| Qwen2-7B | 3.78 | 6.08 | 5.50 | 4.98 | 2.17 | 2.50 | 2.17 | 2.25 | 0.54 | 0.92 | 1.00 | 0.83 | 2.74 | 3.91 | 4.79 | 3.58 | 2.95 |
| Qwen2.5-3B | 2.46 | 3.30 | 6.88 | 3.86 | 1.16 | 1.70 | 2.50 | 1.68 | 0.28 | 0.38 | 0.67 | 0.48 | 10.44 | 13.44 | 19.55 | 14.07 | 5.17 |
| Qwen2.5-7B | 3.23 | 5.24 | 4.91 | 4.48 | 5.60 | 6.08 | 7.92 | 6.45 | 0.37 | 0.58 | 0.58 | 0.58 | 30.30 | 36.80 | 45.18 | 37.31 | 12.89 |
| llava-ov-1B | 3.84 | 4.58 | 12.55 | 6.31 | 3.80 | 3.93 | 13.46 | 5.89 | 1.18 | 1.50 | 2.92 | 1.93 | 19.34 | 20.39 | 38.08 | 24.59 | 9.53 |
| llava-ov-7B | 0.13 | 0.25 | 0.17 | 0.20 | 0.08 | 0.10 | 0.25 | 0.14 | 0.00 | 0.00 | 0.00 | 0.00 | 0.60 | 0.63 | 0.75 | 0.67 | 0.25 |
| InternVL2-1B | 3.81 | 3.75 | 22.88 | 5.53 | 0.47 | 0.45 | 6.00 | 0.84 | 2.27 | 2.13 | 16.73 | 3.66 | 21.08 | 22.87 | 55.22 | 28.11 | 9.54 |
| InternVL2-8B | 0.23 | 0.23 | 2.25 | 0.42 | 2.07 | 2.25 | 7.58 | 2.68 | 0.20 | 0.22 | 0.71 | 0.32 | 2.59 | 2.67 | 5.00 | 3.23 | 1.97 |
| InternVL2.5-1B | 2.73 | 2.73 | 18.58 | 4.63 | 0.34 | 0.34 | 4.67 | 0.61 | 0.15 | 0.17 | 1.38 | 0.29 | 10.39 | 14.43 | 24.47 | 14.25 | 5.49 |
| InternVL2.5-8B | 3.31 | 3.88 | 20.46 | 5.50 | 2.29 | 2.54 | 10.25 | 10.25 | 0.80 | 0.99 | 5.21 | 5.21 | 24.43 | 35.36 | 37.67 | 30.39 | 12.96 |
| GPT-4o | 30.23 | 37.86 | 36.58 | 35.45 | 21.35 | 24.38 | 26.33 | 23.88 | 6.20 | 10.69 | 8.62 | 8.33 | 59.81 | 79.57 | 63.69 | 67.24 | 33.76 |
| Gemini2.0-Flash | 8.43 | 10.45 | 18.25 | 11.35 | 10.84 | 12.08 | 13.00 | 11.79 | 5.47 | 7.48 | 15.08 | 7.91 | 16.76 | 20.33 | 19.54 | 18.67 | 12.98 |
| Gemini2.5-Flash | 30.23 | 36.08 | 37.13 | 35.02 | 30.73 | 37.00 | 34.00 | 33.89 | 16.79 | 22.55 | 24.70 | 21.39 | 55.83 | 66.71 | 60.39 | 60.98 | 34.73 |
| **Ours (Embedding)** | **39.97** | **57.26** | **52.89** | **49.98** | **37.82** | **40.09** | **72.96** | **48.42** | **25.58** | **28.68** | **49.14** | **32.93** | **70.01** | **86.38** | **75.94** | **77.32** | **52.84** |
| GPT-3.5-turbo (Without Image) | 2.58 | 3.03 | 3.30 | 3.02 | 2.45 | 2.53 | 3.83 | 2.68 | 0.00 | 0.00 | 0.00 | 0.00 | 2.44 | 3.38 | 2.56 | 2.73 | 2.16 |
| GPT-4 (Without Image) | 0.43 | 0.75 | 0.45 | 0.56 | 0.08 | 0.11 | 0.67 | 0.15 | 0.01 | 0.01 | 0.52 | 0.01 | 0.00 | 0.00 | 0.00 | 0.00 | 0.23 |
| GPT-4o (Without Image) | 16.70 | 22.54 | 19.01 | 19.80 | 11.41 | 11.64 | 13.63 | 13.02 | 3.41 | 4.58 | 5.90 | 4.09 | 27.56 | 46.45 | 28.67 | 32.60 | 17.66 |

Table 4: **Evaluation Setting 3.** IoU, P, R, F1 represent bounding-box-level IoU score, precision, recall and F1 score. Avg represents the average score on all datasets and evaluation metrics and the ordering is decided by this score.

| Settings | Evaluation Setting 1 | | | | | Evaluation Setting 2 | | | | | Evaluation Setting 3 | | | | | Avg |
|---|---|---|---|---|---|---|---|---|---|---|---|---|---|---|---|---|
| Testsets | Chart | Doc | Info | Trins | Avg | Chart | Doc | Info | Trins | Avg | Chart | Doc | Info | Trins | Avg | Avg |
| Idefics2-8b | 9.50 | 8.50 | 3.00 | 0.00 | 5.25 | 0.00 | 6.50 | 18.50 | 3.50 | 7.13 | 0.50 | 1.50 | 5.50 | 0.00 | 1.88 | 4.75 |
| LLaVa-7b-16 | 0.00 | 0.00 | 0.00 | 0.00 | 0.00 | 0.00 | 0.00 | 0.00 | 0.00 | 0.00 | 15.50 | 19.50 | 9.00 | 14.00 | 14.50 | 4.83 |
| Deepseek-VL-7b-chat | 23.50 | 2.50 | 15.50 | 0.00 | 10.38 | 4.00 | 3.00 | 0.00 | 3.00 | 2.50 | 3.50 | 3.00 | 2.00 | 0.00 | 2.13 | 5.00 |
| InternLM-XP2-4k-7b | 6.50 | 3.00 | 13.00 | 3.00 | 6.38 | 3.50 | 0.50 | 3.50 | 10.50 | 4.50 | 5.50 | 1.00 | 9.50 | 8.50 | 6.00 | 5.63 |
| Phi3V | 7.50 | 4.50 | 2.50 | 0.00 | 3.63 | 15.00 | 7.00 | 7.50 | 6.00 | 8.88 | 17.50 | 2.50 | 8.00 | 13.00 | 10.25 | 7.58 |
| LLaVa-13b-16 | 0.00 | 0.00 | 0.00 | 0.00 | 0.00 | 0.00 | 0.00 | 0.00 | 0.00 | 0.00 | 57.00 | 12.50 | 13.00 | 48.50 | 32.75 | 10.92 |
| CogVLM2-Llama3-19b | 24.00 | 5.00 | 8.00 | 6.00 | 10.75 | 7.50 | 10.50 | 12.00 | 9.50 | 9.88 | 8.50 | 10.00 | 19.50 | 12.00 | 12.50 | 11.04 |
| Qwen-VL | 71.50 | 11.00 | 41.00 | 4.50 | 32.00 | 34.50 | 16.00 | 19.00 | 5.50 | 18.75 | 39.00 | 14.00 | 17.50 | 2.50 | 18.25 | 23.00 |
| Monkey-Chat | 47.00 | 38.00 | 54.50 | 12.00 | 37.88 | 45.00 | 12.00 | 10.50 | 0.00 | 16.88 | 52.00 | 29.50 | 17.50 | 1.00 | 25.00 | 26.58 |
| InternLM-XP2-VL-7b | 23.00 | 21.00 | 22.50 | 10.00 | 19.13 | 26.00 | 27.50 | 53.50 | 20.50 | 31.88 | 25.50 | 28.00 | 43.00 | 24.50 | 30.25 | 27.08 |
| MiniCPM-Llama3 | 53.00 | 58.00 | 73.00 | 50.50 | 58.63 | 29.50 | 88.00 | 88.00 | 70.50 | 69.00 | 36.00 | 81.00 | 83.50 | 80.00 | 70.13 | 55.54 |
| GPT4o | 98.00 | 94.00 | 94.00 | 98.00 | 96.00 | 100.00 | 99.00 | 100.00 | 100.00 | 99.75 | 100.00 | 100.00 | 99.00 | 100.00 | 99.75 | 98.50 |
| GPT-3.5-turbo (w/o Image) | - | - | - | - | - | 20.00 | 48.00 | 38.50 | 21.00 | 31.88 | 20.50 | 34.00 | 23.00 | 9.50 | 21.75 | 26.81 |
| GPT-4 (w/o Image) | - | - | - | - | - | 69.50 | 67.50 | 63.00 | 78.50 | 69.63 | 2.00 | 2.50 | 12.50 | 0.00 | 4.25 | 36.94 |
| GPT-4o (w/o Image) | - | - | - | - | - | 84.00 | 93.50 | 92.00 | 82.50 | 88.00 | 92.00 | 98.00 | 98.50 | 93.00 | 95.38 | 91.69 |

Table 5: **The Instruction-Following Rate.** This value represents VLMs' capability to follow instructions for this task, i.e., generate at least one bounding box regardless of the correctness of the generation. Avg represents the average score on all datasets and settings, and the ordering is decided by this score.

circumstances, the testing model still needs to have a spatial understanding to select the correct grounded bounding boxes with no need to generate coordinates from scratch. Though this setting is not aligned with the most common practical scenarios, it serves as a reasonable evaluation setting. The evaluation metrics used for this setting are the same as in setting 2.

# F MORE EXPERIMENTAL RESULTS

## F.1 PERFORMANCE ON EVALUATION SETTING 3

The performance on evaluation setting 3 is shown in Table 6.

## F.2 PERFORMANCE ON INSTRUCTION-FOLLOWING RATE

The results of the instruction-following rates are shown in Table 5.

## F.3 FURTHER FINDINGS

> **Finding 3.** The instruction-following rates are not positively correlated to the difficulty of each evaluation setting.

In qualitative analysis, evaluation setting 2 is easier than evaluation setting 3 as it provides more OCR information, making it possible to directly answer the question without a real spatial understanding,

| Metrics | ChartQA | | | | DocVQA | | | | InfographicsVQA | | | | TRINS | | | | Avg |
|---|---|---|---|---|---|---|---|---|---|---|---|---|---|---|---|---|---|
| Testsets | IoU | P | R | F1 | IoU | P | R | F1 | IoU | P | R | F1 | IoU | P | R | F1 | Avg |
| Idefics2-8b | 0.00 | 0.00 | 0.00 | 0.00 | 0.00 | 0.00 | 0.00 | 0.00 | 0.00 | 0.00 | 0.00 | 0.00 | 0.00 | 0.00 | 0.00 | 0.00 | 0.00 |
| DeepSeek-VL-7B-chat | 0.38 | 0.38 | 3.00 | 0.67 | 0.08 | 0.08 | 1.00 | 0.16 | 0.03 | 0.03 | 0.13 | 0.05 | 0.00 | 0.00 | 0.00 | 0.00 | 0.37 |
| InternLM-XComposer2-4KHD-7B | 0.23 | 0.26 | 0.83 | 0.36 | 0.00 | 0.00 | 0.00 | 0.00 | 0.22 | 0.26 | 2.42 | 0.38 | 2.13 | 5.13 | 2.43 | 2.97 | 1.10 |
| CogVLM2-Llama3-19B | 0.36 | 0.37 | 3.17 | 0.64 | 0.31 | 0.31 | 3.50 | 0.55 | 0.42 | 0.46 | 2.50 | 0.73 | 3.54 | 4.60 | 7.27 | 4.67 | 2.09 |
| Phi3-V | 1.12 | 1.36 | 5.92 | 1.81 | 0.69 | 0.69 | 1.50 | 0.81 | 0.32 | 0.45 | 2.29 | 0.56 | 6.05 | 6.38 | 11.50 | 7.17 | 3.04 |
| LLaVA-v1.6-Vicuna-7B | 0.78 | 0.93 | 5.56 | 1.35 | 0.77 | 0.75 | 9.33 | 1.35 | 0.50 | 0.52 | 3.20 | 0.88 | 3.55 | 3.56 | 11.83 | 5.08 | 3.12 |
| InternLM-XComposer2-VL-7B | 2.03 | 3.51 | 5.29 | 3.00 | 2.35 | 3.00 | 2.50 | 2.67 | 0.48 | 1.14 | 2.83 | 0.77 | 8.70 | 12.93 | 10.10 | 10.26 | 4.47 |
| Qwen-VL | 4.25 | 4.21 | 35.82 | 7.35 | 1.14 | 1.13 | 11.75 | 2.02 | 0.80 | 0.82 | 7.60 | 1.46 | 0.89 | 0.89 | 2.50 | 1.20 | 5.24 |
| Monkey-chat | 6.15 | 6.46 | 40.06 | 10.30 | 1.51 | 1.50 | 18.58 | 2.72 | 1.25 | 1.26 | 11.92 | 2.25 | 0.70 | 0.70 | 1.00 | 0.79 | 6.70 |
| MiniCPM-Llama3-V 2.5 | 2.40 | 3.61 | 3.92 | 3.04 | 4.86 | 6.02 | 5.83 | 5.21 | 0.48 | 0.81 | 1.63 | 0.70 | 23.82 | 41.41 | 27.08 | 29.46 | 10.02 |
| LLaVA-v1.6-Vicuna-13B | 6.21 | 6.49 | 35.55 | 10.01 | 1.72 | 1.72 | 6.00 | 2.20 | 0.76 | 0.74 | 5.11 | 1.26 | 15.58 | 16.54 | 39.08 | 20.85 | 10.61 |
| GPT-4o | 30.23 | 37.86 | 36.58 | 35.45 | 21.35 | 24.38 | 26.33 | 23.88 | 6.20 | 10.69 | 8.62 | 8.33 | 59.81 | 79.57 | 63.69 | 67.24 | 33.76 |
| GPT-3.5-turbo (Without Image) | 2.58 | 3.03 | 3.30 | 3.02 | 2.45 | 2.53 | 3.83 | 2.68 | 0.00 | 0.00 | 0.00 | 0.00 | 2.44 | 3.38 | 2.56 | 2.73 | 2.16 |
| GPT-4 (Without Image) | 0.43 | 0.75 | 0.45 | 0.56 | 0.08 | 0.11 | 0.67 | 0.15 | 0.01 | 0.01 | 0.52 | 0.01 | 0.00 | 0.00 | 0.00 | 0.00 | 0.23 |
| GPT-4o (Without Image) | 16.70 | 22.54 | 19.01 | 19.80 | 11.41 | 11.64 | 13.63 | 13.02 | 3.41 | 4.58 | 5.90 | 4.09 | 27.56 | 46.45 | 28.67 | 32.60 | 17.66 |

Table 6: **Evaluation Setting 3.** IoU, P, R, F1 represent bounding-box-level IoU score, precision, recall and F1 score. Avg represents the average score on all datasets and evaluation metrics and the ordering is decided by this score.

which is also verified by GPT4o's performance gap. However, things are different when it comes to open-source MLLMs. The open-source MLLMs' average performance on evaluation setting 2, shown in Table 2, is abnormally lower than setting 3, shown in Table 6, which is contradictory to their natural difficulties. By further inspection, we find this phenomenon is caused by the relatively lower average instruction-following rate in evaluation setting 2. This reveals that the provision of more information, i.e., the OCR texts, does harm to MLLMs' ability to correctly follow instructions, thus leading to lower performances. This finding reveals not only the inability of existing open-source MLLMs to follow complex instructions but also their poor robustness: even the useful information might harm the MLLMs' instruction-following ability.

# G   DETAILED ABLATION STUDY

This section mainly focuses on the ablation experiments toward our embedding-based methods on the OCR-free Grounding setting since it directly measures the correctness of selected image patches for grounding.

**Effect of Embedding Merging.** We evaluate the effectiveness of utilizing our proposed embedding merging techniques. As shown in Table 7, without embedding merging, the average performance is the lowest among the settings in which the embedding merging is utilized. In the table, $3 \times 3$ Similarity Merging represents merging the similarities within the $3 \times 3$ surrounding window of each patch. Among all the experimental settings, the utilization of $3 \times 3$ embedding merging reaches the best performance. The results are reasonable since embedding merging with a smaller window size can alleviate the randomness, while a larger window might smooth the values too much, causing the similarities to be too close to distinguish.

**Effect of 2-Level Selection.** As shown in Table 8, the settings notated as "Top-k Patches" represent directly selecting the

| Testsets | Chart | Doc | Info | Trins | Avg |
|---|---|---|---|---|---|
| Metrics (%) | IoU | IoU | IoU | IoU | Avg |
| No Similarity Merging | 8.93 | 9.27 | 6.72 | 14.14 | 9.77 |
| $3 \times 3$  Similarity Merging | 10.17 | 11.87 | 7.66 | 15.65 | **11.34** |
| $5 \times 5$  Similarity Merging | 10.01 | 11.32 | 7.32 | 15.14 | 10.95 |
| $7 \times 7$  Similarity Merging | 9.47 | 10.87 | 6.89 | 14.76 | 10.50 |
| Top-5 Patches | 9.95 | 12.48 | 7.11 | 6.56 | 9.02 |
| Top-10 Patches | 10.37 | 14.10 | 8.16 | 9.42 | 10.51 |
| Top-15 Patches | 9.92 | 13.77 | 7.56 | 11.25 | 10.63 |
| Top-20 Patches | 9.50 | 12.75 | 6.98 | 12.42 | 10.41 |
| Top-25 Patches | 8.95 | 12.05 | 6.50 | 13.22 | 10.18 |
| Top-30 Patches | 8.45 | 11.19 | 6.00 | 13.91 | 9.89 |
| Top-5 + Top-10 Patches | 10.88 | 14.90 | 8.53 | 9.34 | 10.91 |
| Top-5 + Top-15 Patches | 10.96 | 15.38 | 8.40 | 11.13 | 11.47 |
| Top-5 + Top-20 Patches | 10.86 | 15.26 | 8.20 | 12.24 | 11.64 |
| Top-5 + Top-25 Patches | 10.70 | 15.24 | 8.06 | 13.13 | 11.78 |
| Top-5 + Top-30 Patches | 10.51 | 15.02 | 7.85 | 13.88 | **11.82** |

Table 7: Ablation studies of our embedding-based method on OCR-free grounding setting. The results illustrate the effectiveness of our proposed embedding merging and 2-level selection mechanism.

top-k patches as the grounded area, while settings notated as "Top-k1 + Top-k2 Patches" represent first selecting the top-k1 patches and then adding the Top-k2 patches gradually if they appear in the surroundings of top-k1 patches. From the results, several findings can be observed: (1) When utilizing only the top-k patches without the 2-level selection, the performance first grows and then starts to decline gradually. This phenomenon indicates that when more patches are selected, the growth of the union is faster than the intersection. (2) When the 2-level selection is utilized, the performance is consistently higher than without using it, and no clear decline trend is observed.

| Testsets | Chart | Doc | Info | Trins | Avg |
|---|---|---|---|---|---|
| **Metrics** (%) | **IoU** | **IoU** | **IoU** | **IoU** | **Avg** |
| Top-5 Patches | 9.95 | 12.48 | 7.11 | 6.56 | 9.02 |
| Top-10 Patches | 10.37 | 14.10 | 8.16 | 9.42 | 10.51 |
| Top-15 Patches | 9.92 | 13.77 | 7.56 | 11.25 | 10.63 |
| Top-20 Patches | 9.50 | 12.75 | 6.98 | 12.42 | 10.41 |
| Top-25 Patches | 8.95 | 12.05 | 6.50 | 13.22 | 10.18 |
| Top-30 Patches | 8.45 | 11.19 | 6.00 | 13.91 | 9.89 |
| Top-35 Patches | 7.90 | 10.37 | 5.59 | 14.51 | 9.60 |
| Top-40 Patches | 7.53 | 9.77 | 5.35 | 15.06 | 9.43 |
| Top-45 Patches | 7.14 | 9.17 | 5.09 | 15.60 | 9.25 |
| Top-50 Patches | 6.82 | 8.62 | 4.82 | **15.94** | 9.05 |
| Top-3 + Top-5 Patches | 9.97 | 12.58 | 7.26 | 6.18 | 9.00 |
| Top-3 + Top-10 Patches | 10.89 | 15.03 | 8.76 | 8.54 | 10.80 |
| Top-3 + Top-15 Patches | **11.07** | 15.64 | **8.86** | 10.06 | 11.41 |
| Top-3 + Top-20 Patches | 11.01 | 15.60 | 8.72 | 11.01 | 11.58 |
| Top-3 + Top-25 Patches | 10.89 | **15.78** | 8.68 | 11.80 | 11.79 |
| Top-3 + Top-30 Patches | 10.70 | 15.60 | 8.47 | 12.47 | 11.81 |
| Top-3 + Top-35 Patches | 10.39 | 15.35 | 8.38 | 13.00 | 11.78 |
| Top-3 + Top-40 Patches | 10.27 | 15.22 | 8.32 | 13.36 | 11.79 |
| Top-3 + Top-45 Patches | 10.13 | 15.03 | 8.32 | 13.84 | 11.81 |
| Top-3 + Top-50 Patches | 9.97 | 14.82 | 8.16 | 14.16 | 11.78 |
| Top-5 + Top-5 Patches | 9.95 | 12.48 | 7.11 | 6.56 | 9.02 |
| Top-5 + Top-10 Patches | 10.88 | 14.90 | 8.53 | 9.34 | 10.91 |
| Top-5 + Top-15 Patches | 10.96 | 15.38 | 8.40 | 11.13 | 11.47 |
| Top-5 + Top-20 Patches | 10.86 | 15.26 | 8.20 | 12.24 | 11.64 |
| Top-5 + Top-25 Patches | 10.70 | 15.24 | 8.06 | 13.13 | 11.78 |
| Top-5 + Top-30 Patches | 10.51 | 15.02 | 7.85 | 13.88 | **11.82** |
| Top-5 + Top-35 Patches | 10.16 | 14.73 | 7.67 | 14.42 | 11.75 |
| Top-5 + Top-40 Patches | 10.00 | 14.57 | 7.65 | 14.91 | 11.79 |
| Top-5 + Top-45 Patches | 9.84 | 14.38 | 7.62 | 15.45 | 11.82 |
| Top-5 + Top-50 Patches | 9.68 | 14.20 | 7.44 | 15.82 | 11.78 |

Table 8: Detailed ablation studies of our embedding-based method on OCR-free grounding setting. The results illustrate the effectiveness of the 2-level selection mechanism.

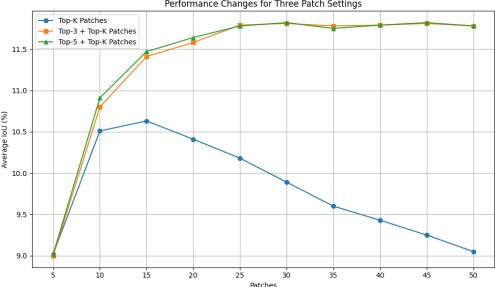

Figure 6: The visualization of detailed ablation studies of our embedding-based method on an OCR-free grounding setting for the 2-level selection mechanism. The utilization of the 2-level selection mechanism avoids the performance decline when more patches are selected, making the performance more stable and resulting in better performance.

This phenomenon indicates that the use of this selection metric can largely alleviate the selection of unimportant patches.

Figure 6 presents the performance changes with or without our 2-level selection mechanism. As shown in the figure, when no 2-level selection mechanism is utilized, the performance declines consistently after $K = 15$, as more random patches are included, making the grounding area unstable. On the contrary, when our 2-level selection mechanism is utilized, only the patches in the surroundings of the top-K1 patches are included, which directly removes the potentially noisy patches. To conclude, the utilization of the 2-level selection mechanism avoids the performance decline when more patches are selected, making the performance more stable and resulting in better performance.

---

Prompt for Grounded Bounding Box **Generation**

---

**System Prompt**
You are a helpful and precise assistant in finding the grounding bounding boxes given the question-answer pair and the poster image.

**User Prompt**
Question: {*question*}
Answer: {*answer*}
Above is the question and answer for a given poster.
Sentence-level bounding boxes with indexes are provided in the poster, and detailed indexes with corresponding texts are also provided below:
{*idx-text pairs*}
Can you provide me with the indices of bounding boxes that can accurately and sufficiently lead to the answer? Make sure you check both the poster image and the text provided above.
Please provide the index in the first line, use a comma to separate different indexes if more than one, and do not output anything else except for indexes or commas.
Please then provide the reason in the following lines on why you chose those bounding boxes.

---

Figure 7: The prompt we used to request GPT4o to generate the grounded bounding boxes that support the answer to the question.

## H  GENERATION & EVALUATION PROMPTS

### H.1  GROUNDING GENERATION PROMPTS

The prompt we use for generating grounded bounding boxes is shown in Figure 7, and the prompt we use for evaluating the correctness of the previously generated bounding boxes.

### H.2  EVALUATION PROMPTS

In order to alleviate the influences of prompts when testing on our benchmark, 3 different prompts with diverse formatting requirements are proposed for each evaluation setting, resulting in a total of 9 different evaluation prompts. For each evaluation setting, we present the best results across the 3 different prompts, as shown in Figure 9. The main difference between different prompts in the same evaluation setting is the required format for the generation or selection of the grounded bounding boxes. Specifically, prompts with "#1" utilize the CSS format, which is widely used for pretraining; prompts with "#2" utilize the most naive format containing the necessary information in a list; prompts with "#3" are similar to "#2" but utilize the relative coordinates.

## I  BENCHMARK DATA EXAMPLES

The benchmark data examples are visualized in Figure 10 for ChatQA, Figure 11 for DocVQA, Figure 12 for InfographicsVQA, and Figure 13 for TRINS. The questions and answers are given in text form, and the supported grounded bounding boxes are directly visualized in the images. The samples from different sources show the diversity of our benchmark data.

---

Prompt for Grounded Bounding Box **Rectification**

---

**System Prompt**
You are a helpful and precise assistant in analyzing the grounding bounding boxes given the question-answer pair and the poster image.

**User Prompt**
Question: {*question*}
Answer: {*answer*}
Above is the question and answer for a given poster.
Sentence-level bounding boxes with indexes are provided in the poster, and detailed indexes with corresponding texts are also provided below:
{*idx-text pairs*}
Do you think the bounding boxes with index {*idx-list*} can accurately and sufficiently lead to the answer to the given question? Please output YES or NO in the first line, then provide the reason in the following lines.

---

Figure 8: The prompt we used to request GPT4o to evaluate the grounded bounding boxes generated in the previous step.

```
Prompt # 1 for Evaluation Setting 1

{question}
Please first answer the question according to the given image and
then generate the grounded text bounding boxes that support your
answer. The width of the image is {width}px and the height is
{height}px. Please generate the bounding boxes in the CSS format:
textElement {{
    left: {left}px;
    top: {top}px;
    width: {width}px;
    height: {height}px;
    content: "{text}";
}}
```

```
Prompt # 2 for Evaluation Setting 1

{question}
Please first answer the question according to the given image and
then generate the grounded text bounding boxes that support your
answer. The width of the image is {width}px and the height is
{height}px. Please generate the bounding boxes in the below
format, the coordinates should be intergers:
[{left}, {top}, {width}, {height}, "{text}"]
```

```
Prompt # 3 for Evaluation Setting 1

{question}
Please first answer the question according to the given image and
then generate the grounded text bounding boxes that support your
answer. Please generate the bounding boxes in the below format
using relative coordinates:
[{left}, {top}, {width}, {height}, "{text}"]
```

```
Prompt # 1 for Evaluation Setting 2

{question}
Please first answer the question according to the given image and
then select the grounded bounding boxes that support your answer.
The width of the image is {width}px and the height is {height}px. All
the bounding boxes are provided below in the CSS format:
textElement-1 {{
    left: {left}px;
    top: {top}px;
    width: {width}px;
    height: {height}px;
    content: "{text}";
}},
textElement-2 {{
    left: {left}px;
    top: {top}px;
    width: {width}px;
    height: {height}px;
    content: "{text}";
}},
...
```

```
Prompt # 2 for Evaluation Setting 2

{question}
Please first answer the question according to the given image and
then select the grounded bounding boxes that support your answer.
The width of the image is {width}px and the height is {height}px. All
the bounding boxes are provided below in the below format::
[1, [{left}, {top}, {width}, {height}, "{text}"]],
[2, [{left}, {top}, {width}, {height}, "{text}"]],
...
```

```
Prompt # 3 for Evaluation Setting 2

{question}
Please first answer the question according to the given image and
then select the grounded bounding boxes that support your answer.
The width of the image is {width}px and the height is {height}px. All
the bounding boxes are provided below in the below format::
[1, [{left}, {top}, {width}, {height}, "{text}"]],
[2, [{left}, {top}, {width}, {height}, "{text}"]],
...
```

```
Prompt # 1 for Evaluation Setting 3

{question}
Please first answer the question according to the given image and
then select the grounded bounding boxes that support your answer.
The width of the image is {width}px and the height is {height}px. All
the bounding boxes are provided below in the CSS format:
textElement-1 {{
    left: {left}px;
    top: {top}px;
    width: {width}px;
    height: {height}px;
}},
textElement-2 {{
    left: {left}px;
    top: {top}px;
    width: {width}px;
    height: {height}px;
}},
...
```

```
Prompt # 2 for Evaluation Setting 3

{question}
Please first answer the question according to the given image and
then select the grounded bounding boxes that support your answer.
The width of the image is {width}px and the height is {height}px. All
the bounding boxes are provided below in the below format::
[1, [{left}, {top}, {width}, {height}]],
[2, [{left}, {top}, {width}, {height}]],
...
```

```
Prompt # 3 for Evaluation Setting 3

{question}
Please first answer the question according to the given image and
then select the grounded bounding boxes that support your answer.
The width of the image is {width}px and the height is {height}px. All
the bounding boxes are provided below in the below format:
[1, [{left}, {top}, {width}, {height}]],
[2, [{left}, {top}, {width}, {height}]],
...
```

Figure 9: The evaluation prompts for different evaluation settings.

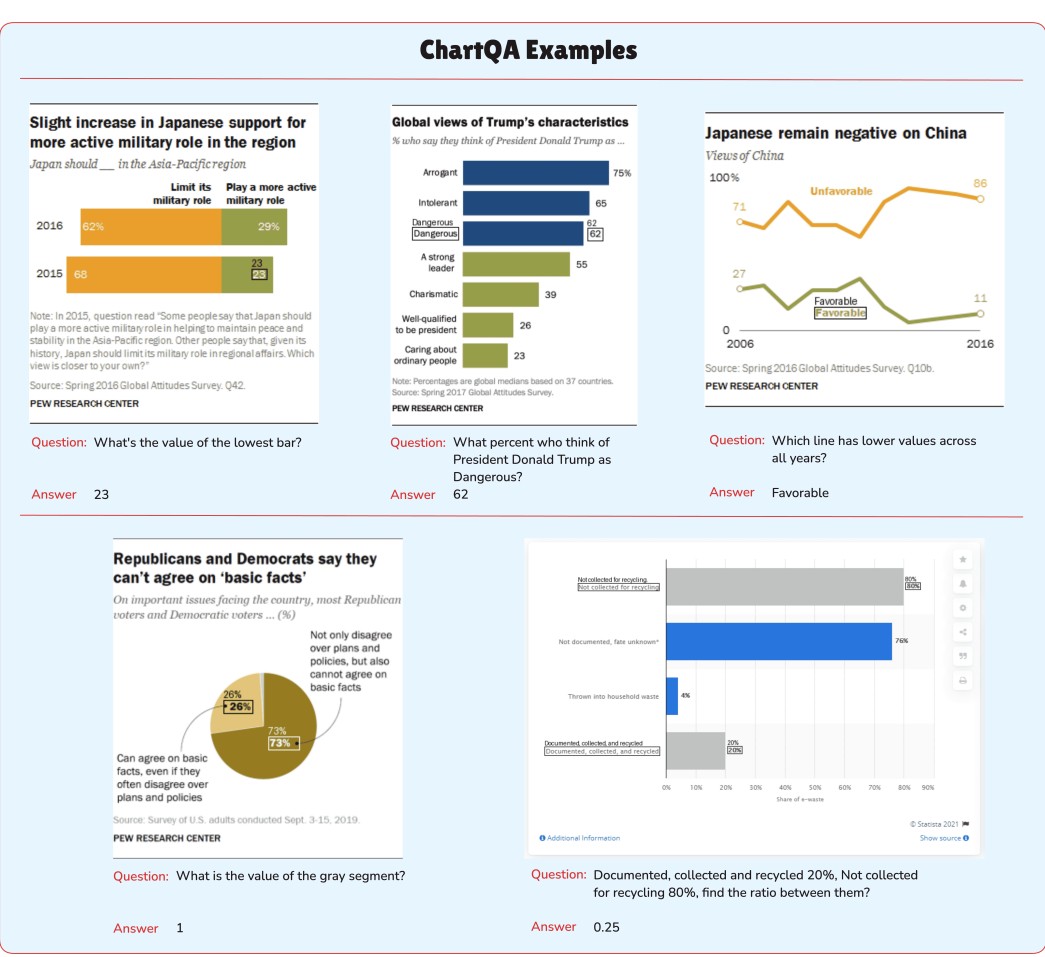

Figure 10: Benchmark data examples from ChatQA. The grounded bounding boxes have already been visualized in the original image for better illustration.

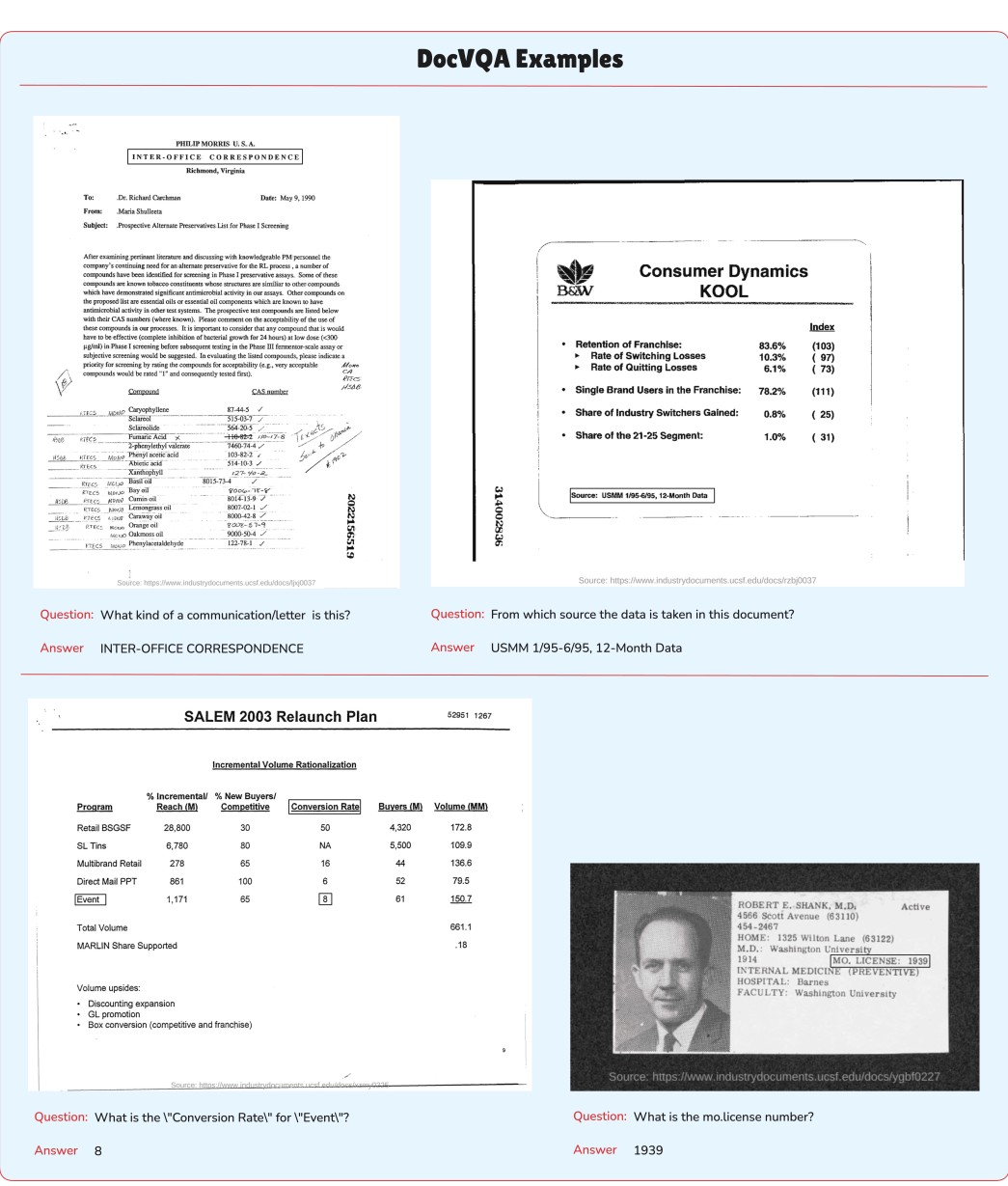

Figure 11: Benchmark data examples from DocVQA. The grounded bounding boxes have already been visualized in the original image for better illustration.

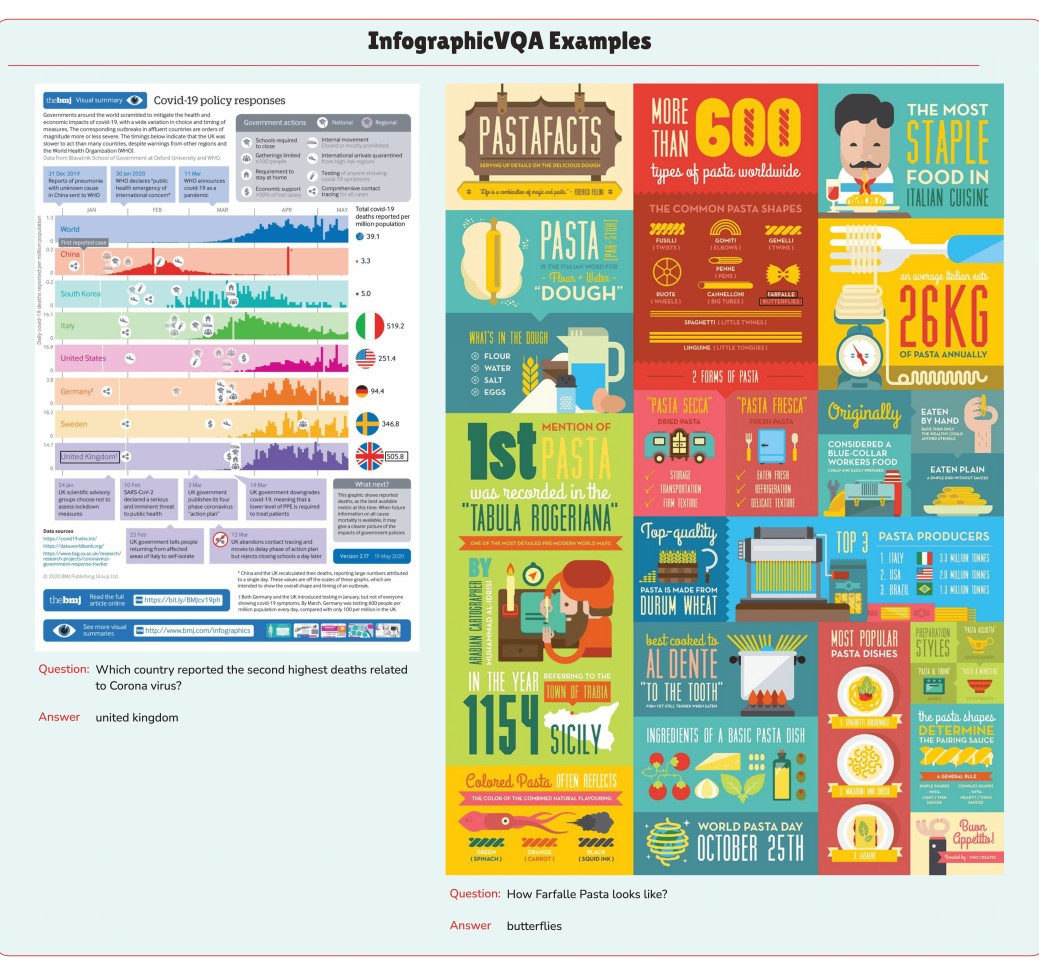

Figure 12: Benchmark data examples from InfographicsVQA. The grounded bounding boxes have already been visualized in the original image for better illustration.

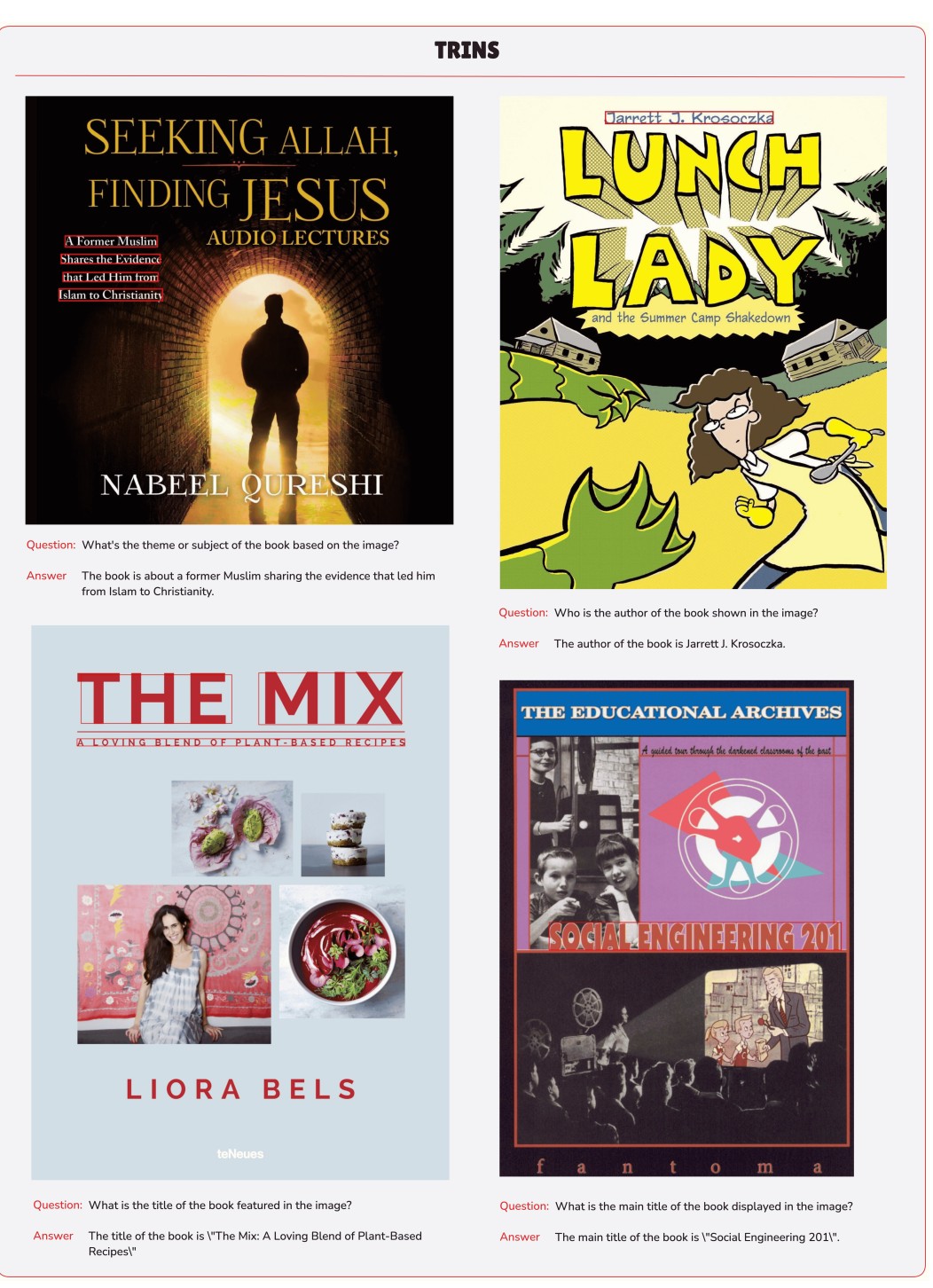

Figure 13: Benchmark data examples from TRINS. The grounded bounding boxes have already been visualized in the original image for better illustration.

