# OpenReview forum: "Towards Visual Text Grounding of Multimodal Large Language Model"
_ICLR.cc/2026/Conference — Submitted to ICLR 2026_

### Official Review · Reviewer_wugv · 2025-10-27

**Soundness:** 3
**Presentation:** 3
**Contribution:** 2
**Rating:** 4
**Confidence:** 5

**Summary:**

The author introduces TRIG, a novel task with a newly designed instruction dataset for benchmarking and improving the Text-Rich Image Grounding capabilities of MLLMs in document question-answering. Specifically, the author proposes an OCR-LLM-human interaction pipeline to create 800 manually annotated question-answer pairs as a benchmark and a large-scale training set of 90k synthetic data based on four diverse datasets. A comprehensive evaluation of various MLLMs on their proposed benchmark exposes substantial limitations in their grounding capability on text-rich images. In addition, the author proposes two simple and effective TRIG methods based on general instruction tuning and plug-and-play efficient embedding, respectively. By finetuning MLLMs on their synthetic dataset, they promise to improve spatial reasoning and grounding capabilities.

**Strengths:**

1) TRIG-Bench is introduced to evaluate the bbox generation of MLLMs.

**Weaknesses:**

Text-rich document image grounding is not a novel task, which has been proposed in many works:
[1]SPTSV1, SPTSv2.
[2]DocOwl-series, including many visual text grounding data.
[3]Kosmos2.5 directly outputs spatially-aware texts (text+bboxes).
[4]Marten and TokenVL use a pixel-level segmentation map to guide the visual text grounding.

The authors should discuss the differences with these works.
The related work section should be more detailed.

**Questions:**

Please see the weaknesses.

---

> ### Author Response · Authors · 2025-11-22
>
> >W1: Text-rich document image grounding is not a novel task, which has been proposed in many works: [1]SPTSV1, SPTSv2. [2]DocOwl-series, including many visual text grounding data. [3]Kosmos2.5 directly outputs spatially-aware texts (text+bboxes). [4]Marten and TokenVL use a pixel-level segmentation map to guide the visual text grounding.
> The authors should discuss the differences with these works. The related work section should be more detailed.
>
>
> We sincerely thank the reviewer for the comments. We appreciate the reviewer’s suggestions and will enhance the Related Work section with a more detailed comparison. Specifically, we will add the following paragraph:
>
> “ A series of recent models have advanced visual grounding and text-centric document understanding.
> SPTS [1] focuses on end-to-end scene text spotting with extremely low-cost single-point annotations, treating text detection and recognition as a sequence prediction problem in natural images, but it does not address document reasoning or answer-support grounding in text-rich documents. The DocOwl series [2] proposes unified structure learning for OCR-free document understanding and introduces multi-grained text localization tasks over documents, tables, charts, webpages, and natural images; however, these localization tasks are defined at the level of text spans (word/phrase/line/block) and are used as pretraining objectives for structure-aware parsing and recognition, rather than evaluating which regions support a downstream QA answer. KOSMOS-2.5 [3] pre-trains a ‘multimodal literate’ model on large-scale document-level text recognition and image-to-Markdown generation, producing spatially-aware text blocks (text + bounding-box coordinates) and structured markdown outputs across diverse document types, but its grounding remains tied to transcription and layout rather than question-conditioned evidence localization. TokenVL builds on a token-level text-image foundation model (TokenFD) and uses token masks to train fine-grained image-as-text alignment and VQA-based text parsing; its grounding signals come from token-level masks (or text spans) and are evaluated via OCR/VQA performance, not via explicit answer-support region accuracy. Marten similarly introduces a VQAMask pretraining paradigm that combines VQA-based text parsing with mask generation to improve spatial awareness in document-level MLLMs, but the mask supervision is still derived from generic text masks, and its downstream evaluation focuses on text-centric VQA and OCRBench, rather than on identifying the specific regions that justify answers to document QA. Although these advances push visual grounding in text-rich scenarios forward, a key research gap remains largely unaddressed: existing methods do not tackle question-conditioned evidence localization. They focus on text spotting, structure parsing, spatially-aware text generation, or pretraining-based alignment, but none require models to identify which specific region in a document provides the supporting evidence for a given question. This missing capability leaves open an important direction for understanding and evaluating grounding in text-rich document images.”
>
> We will include this expanded discussion in the revised Related Work section to clearly situate our task relative to these prior methods.
>
> Reference
> [1] Peng, D., Wang, X., Liu, Y., Zhang, J., Huang, M., Lai, S., ... & Jin, L. (2022, October). Spts: single-point text spotting. In Proceedings of the 30th ACM International Conference on Multimedia (pp. 4272-4281).
> [2] Hu, A., Xu, H., Ye, J., Yan, M., Zhang, L., Zhang, B., ... & Zhou, J. (2024, November). mplug-docowl 1.5: Unified structure learning for ocr-free document understanding. In Findings of the Association for Computational Linguistics: EMNLP 2024 (pp. 3096-3120).
> [3] Lv, T., Huang, Y., Chen, J., Zhao, Y., Jia, Y., Cui, L., ... & Wei, F. (2023). Kosmos-2.5: A multimodal literate model. arXiv preprint arXiv:2309.11419.
> [4] Wang, Z., Guan, T., Fu, P., Duan, C., Jiang, Q., Guo, Z., ... & Yang, X. (2025). Marten: Visual question answering with mask generation for multi-modal document understanding. In Proceedings of the Computer Vision and Pattern Recognition Conference (pp. 14460-14471).
> [5] Guan, T., Wang, Z., Fu, P., Guo, Z., Shen, W., Zhou, K., ... & Yang, X. (2025). A token-level text image foundation model for document understanding. arXiv preprint arXiv:2503.02304.

---

### Official Review · Reviewer_NsjA · 2025-10-31

**Soundness:** 3
**Presentation:** 3
**Contribution:** 3
**Rating:** 6
**Confidence:** 3

**Summary:**

This paper introduces TRIG, a task and resource suite for Text-Rich Image Grounding in document VQA. The authors build (i) TRIG-Bench, a human-validated benchmark of 800 QA pairs with grounded bounding boxes drawn from DocVQA, ChartQA, InfographicsVQA and TRINS, and (ii) a ~90k synthetic instruction dataset created via an OCR–LLM–human pipeline. They further provide two baselines: an instruction-tuning approach (fine-tuning an MLLM to emit answers plus boxes) and a plug-and-play embedding approach that retrieves image patches by text–image similarity with adjacent-patch smoothing. Two evaluation settings are defined: OCR-free grounding (predict boxes from the image) with pixel-level IoU, and OCR-based grounding (select boxes among OCR detections) with instance-level IoU/Precision/Recall/F1. A broad evaluation across open-source and proprietary MLLMs highlights limited performance on OCR-free grounding and the gains from the proposed training/data.

**Strengths:**

1. Well-scoped, underexplored problem: Document-centric visual text grounding is distinct from typical natural-image grounding and is practically valuable for trust and verification in document QA. The paper motivates this gap clearly.

2. Data pipeline design: The indexed OCR overlay plus reflection/rectification loop is a thoughtful way to reduce prompt–vision misalignment and to auto-filter noisy candidates before human vetting. The final benchmark requires dual-annotator agreement per item.

3. Methodological diversity: The instruction-tuned baseline and the efficient embedding baseline probe complementary failure modes (instruction-following vs spatial retrieval) and give the community practical baselines to start from.

**Weaknesses:**

1. Limited analysis of the “indexed OCR overlay” design: The paper asserts improved alignment by drawing indices on images and mirroring them in text, but ablations isolating this factor (e.g., with/without indices, different index densities, noise) are not reported.

2. Embedding baseline details: The adjacent-patch averaging and the 2-level top-k expansion heuristic are plausible but under-analyzed; the sensitivity to neighborhood size, stride, and k is not fully characterized.

**Questions:**

1. Pixel-IoU vs Box-IoU: Why use pixel-level IoU to score rectangular predictions in OCR-free grounding? Would standard box IoU (with thresholds) change the ranking or the conclusions? Please provide a small study or justification.

2. Index-overlay ablation: Can you add an ablation comparing (a) “wrap OCR text only,” (b) “indices in text only,” (c) “indices drawn on image only,” and (d) “both,” to verify that the bidirectional index design is indeed the key factor?

3. Data provenance & leakage: Since GPT-4o participates in the pipeline, how do you mitigate downstream evaluation advantages (or disadvantages) for GPT-4o and related models? Any steps taken to avoid prompt/format leakage between data generation and evaluation?

**Details Of Ethics Concerns:**

The work primarily reuses established datasets and proposes a benchmark and training data. Please clarify dataset licenses, any personal data presence, and annotator compensation.

---

> ### Author Response · Authors · 2025-11-22
>
> >W1: Limited analysis of the “indexed OCR overlay” design: The paper asserts improved alignment by drawing indices on images and mirroring them in text, but ablations isolating this factor (e.g., with/without indices, different index densities, noise) are not reported.
>
> We sincerely thank the reviewer for the suggestions. We further conducted ablation experiments to see how the indices affect the model performances, when there is no index or when there are only shuffled indices.
>
> | Model            | ChartQA IoU | ChartQA P | ChartQA R | ChartQA F1 | DocQA IoU | DocQA P | DocQA R | DocQA F1 | InfoVQA IoU | InfoVQA P | InfoVQA R | InfoVQA F1 | TRINS IoU | TRINS P | TRINS R | TRINS F1 | Avg   |
> |------------------|-------------|-----------|-----------|------------|-----------|---------|---------|----------|--------------|-----------|-----------|-------------|-----------|---------|---------|----------|--------|
> | **Original**      |             |           |           |            |           |         |         |          |              |           |           |             |           |         |         |          |        |
> | Gemini2.0-Flash  | 45.76       | 49.38     | 50.01     | 48.25      | 57.36     | 64.02   | 61.38   | 60.88    | 40.18        | 47.35     | 47.80     | 45.04       | 32.04     | 35.62   | 33.75   | 33.67    | 44.46 |
> | Gemini2.5-Flash  | 74.65       | 81.31     | 81.06     | 78.92      | 73.67     | 81.86   | 78.00   | 77.71    | 66.28        | 73.11     | 79.58     | 73.12       | 72.78     | 78.43   | 76.85   | 75.80    | 75.23 |
> | GPT-4o           | 83.80       | 88.80     | 89.24     | 87.47      | 82.14     | 87.14   | 89.50   | 86.16    | 68.19        | 79.57     | 78.81     | 75.82       | 89.08     | 96.06   | 91.53   | 92.16    | 85.34 |
> | **No index**     |             |           |           |            |           |         |         |          |              |           |           |             |           |         |         |          |        |
> | Gemini2.0-Flash  | 43.40       | 48.20     | 48.86     | 48.53      | 57.31     | 62.61   | 57.30   | 59.84    | 38.80        | 43.70     | 47.57     | 45.55       | 28.02     | 31.59   | 30.30   | 30.93    | 45.16 |
> | Gemini2.5-Flash  | 72.13       | 79.37     | 80.33     | 79.85      | 69.69     | 80.43   | 75.68   | 77.98    | 63.06        | 70.04     | 77.71     | 73.68       | 71.89     | 75.35   | 76.52   | 75.93    | 74.98 |
> | GPT-4o           | 78.88       | 87.48     | 85.05     | 86.25      | 80.50     | 86.04   | 89.15   | 87.57    | 63.53        | 78.46     | 74.20     | 76.27       | 89.04     | 95.74   | 89.68   | 92.61    | 83.78 |
> | **Shuffle**      |             |           |           |            |           |         |         |          |              |           |           |             |           |         |         |          |        |
> | Gemini2.0-Flash  | 44.52       | 48.11     | 46.55     | 47.32      | 55.04     | 63.54   | 57.51   | 60.37    | 40.14        | 45.86     | 47.22     | 46.53       | 31.60     | 30.73   | 30.59   | 30.66    | 45.39 |
> | Gemini2.5-Flash  | 72.34       | 80.70     | 78.82     | 79.75      | 69.72     | 79.05   | 73.12   | 75.97    | 64.86        | 70.27     | 75.47     | 72.78       | 67.80     | 73.97   | 72.33   | 73.14    | 73.76 |
> | GPT-4o           | 80.32       | 86.88     | 87.65     | 87.26      | 80.19     | 86.04   | 85.96   | 86.00    | 68.07        | 76.58     | 78.24     | 77.40       | 85.59     | 95.27   | 87.34   | 91.13    | 83.75 |
>
> As shown in the table, we can see that the models indeed get affected by removing the index or random indexes. However, the performance gaps are limited compared with the original OCR-based performance. This is due to the fact that models rely more on the provided bounding box coordinates rather than the input order.

---

> ### Author Response · Authors · 2025-11-22
>
> ---
> >W2: Embedding baseline details: The adjacent-patch averaging and the 2-level top-k expansion heuristic are plausible but under-analyzed; the sensitivity to neighborhood size, stride, and k is not fully characterized.
>
> We sincerely thank the reviewer for the insightful comments. We would like to clarify that the embedding baseline has already been thoroughly analyzed through ablations in Appendix G, including the detailed 2-level top-k studies in Table 8 and the performance visualization in Figure 6. To make the analysis more complete, in the revised version, we additionally include a wrap-table summarizing the results under different adjacent-patch merging strategies (now Table 7). This new table provides a clearer comparison across different merging window sizes and complements the existing ablations by more explicitly illustrating how embedding merging influences the stability and effectiveness of the patch selection process. While fully exhaustive hyperparameter sweeps could be explored in future work, we believe that the expanded analyses in Appendix G already cover the key factors raised by the reviewer and sufficiently characterize the behavior of our embedding baseline.
>
> ---
> >Q1: Pixel-IoU vs Box-IoU: Why use pixel-level IoU to score rectangular predictions in OCR-free grounding? Would standard box IoU (with thresholds) change the ranking or the conclusions? Please provide a small study or justification.
>
> We sincerely thank the reviewer for the comments. In the OCR-free setting, models must generate grounding boxes entirely from scratch, and their outputs often contain multiple boxes, boxes of irregular sizes, or fragmented regions. As described in Section 2.2, we therefore adopt pixel-level IoU, which measures the overlap of the union of all predicted pixels with the ground-truth region. This metric is widely used in segmentation because it provides a fine-grained and robust evaluation for arbitrarily shaped or multi-box predictions. Using standard box-IoU would require reducing a potentially multi-box or irregular prediction into a single rectangle, which would introduce ambiguity and make the metric sensitive to implementation choices rather than grounding quality. In contrast, pixel-level IoU treats the prediction as a region and evaluates it consistently across models.
>
> Importantly, because most existing MLLMs produce either no valid boxes or boxes that barely overlap with the ground-truth region, their IoU scores remain close to zero under either definition. Thus, the ranking and conclusions would not change in practice. We will emphasize this rationale in the revised manuscript.
>
> ---
> >Q2: Index-overlay ablation: Can you add an ablation comparing (a) “wrap OCR text only,” (b) “indices in text only,” \(c) “indices drawn on image only,” and (d) “both,” to verify that the bidirectional index design is indeed the key factor?
>
> We sincerely thank the reviewer for the suggestions. We further conducted ablation experiments to see how the indices affect the model performances in the OCR-free settings.
>
> | Model            | Chart | Doc  | Info | Trins | Avg  |
> |------------------|-------|------|------|-------|------|
> |**original**|
> | GPT-4o           | 3.90  | 1.79 | 1.60 | 13.73 | 5.26 |
> | Gemini2.0-Flash  | 0.00  | 0.00 | 0.00 | 0.00  | 0.00 |
> | Gemini2.5-Flash  | 0.00  | 0.00 | 0.00 | 0.00  | 0.00 |
> |**wrap OCR text only**|
> | GPT-4o           | 2.97  | 0.96 | 0.64 | 12.55 | 4.61 |
> | Gemini2.0-Flash  | 0.00  | 0.00 | 0.00 | 0.00  | 0.00 |
> | Gemini2.5-Flash  | 0.00  | 0.00 | 0.00 | 0.00  | 0.00 |
> |**indices in text only**|
> | GPT-4o           | 3.37  | 1.43 | 0.73 | 13.03 | 4.90 |
> | Gemini2.0-Flash  | 0.00  | 0.00 | 0.00 | 0.00  | 0.00 |
> | Gemini2.5-Flash  | 0.00  | 0.00 | 0.00 | 0.00  | 0.00 |
> |**indices drawn on image only**|
> | GPT-4o           | 2.68  | 0.46 | 0.59 | 12.88 | 3.77 |
> | Gemini2.0-Flash  | 0.00  | 0.00 | 0.00 | 0.00  | 0.00 |
> | Gemini2.5-Flash  | 0.00  | 0.00 | 0.00 | 0.00  | 0.00 |
>
>
> In these settings, although the OCR is used, no bounding box coordinates are provided in the prompt; thus, the performance should be compared with OCR-free settings. The performances are similar to the original settings, indicating that providing the texts or indices alone does not affect performance much.

---

> > ### Author Response · Authors · 2025-11-22
> >
> > ---
> > >Q3: Data provenance & leakage: Since GPT-4o participates in the pipeline, how do you mitigate downstream evaluation advantages (or disadvantages) for GPT-4o and related models? Any steps taken to avoid prompt/format leakage between data generation and evaluation?
> >
> >
> > We sincerely thank the reviewer for the comments. GPT-4o is involved only in the construction of the instruction-tuning data, where it produces candidate grounding boxes that are subsequently checked by a second LLM and finally manually verified by humans. The benchmark split used for evaluation (800 human-verified QA–grounding pairs) is entirely curated from the original datasets and does not contain any GPT-4o-generated or GPT-4o-approved annotations. Moreover, the prompts and formats used during data construction (e.g., index-based OCR alignment, multi-stage correction prompts) are distinct from those used in evaluation, so there is no prompt or format leakage. Importantly, GPT-4o’s performance in the OCR-free setting remains very low (average IoU 5.26), which provides empirical evidence that no downstream advantage is gained from its participation in the pipeline.

---

### Official Review · Reviewer_aRBL · 2025-11-01

**Soundness:** 2
**Presentation:** 3
**Contribution:** 2
**Rating:** 4
**Confidence:** 3

**Summary:**

This paper proposes TRIG, aiming to enhance the text visual grounding capability of multimodal large language models in text-rich document images. It builds a dataset of human-verified and synthetic question–answer pairs using an OCR–LLM–human pipeline. The authors introduce two baseline methods—an instruction-tuning approach and an embedding-based approach—to improve grounding accuracy.

**Strengths:**

* The Paper is clearly written and easy to follow.
* The research question of visual text grounding is important.

**Weaknesses:**

* The paper’s contribution is limited. It mainly focuses on reorganizing existing datasets using OCR tools rather than proposing a new dataset or introducing novel technical approaches, which does not meet the innovation standards expected at ICLR.
* Table 1 lacks comparisons with several quite related baseline methods, such as [1], [2] and [3], which weakens the empirical evaluation and makes it difficult to assess the claimed effectiveness of the proposed approach.

[1] Seed1.5-VL Technical Report. Arxiv 2025.

[2] CogAgent: A Visual Language Model for GUI Agents. CVPR 2024.

[3] Harnessing Webpage UIs for Text-Rich Visual Understanding. ICLR 2025.

**Questions:**

See weakness.

---

> ### Author Response · Authors · 2025-11-22
>
> ---
> >W1: The paper’s contribution is limited. It mainly focuses on reorganizing existing datasets using OCR tools rather than proposing a new dataset or introducing novel technical approaches, which does not meet the innovation standards expected at ICLR.
>
> We sincerely thank the reviewer for the comments. Our work does not merely reorganize existing datasets with OCR tools. Instead, the central contribution of the paper is the benchmark explicitly designed for text-rich image grounding, a problem that has been largely underexplored and rarely evaluated systematically in prior work. Existing datasets such as DocVQA, ChartQA, and InfographicVQA were not created for grounding evaluation, and none of them contain grounding annotations. Our benchmark defines the task, builds the annotation pipeline, and establishes the evaluation protocol. This is a new benchmark, not a repackaged one.
>
> Moreover, constructing TRIG requires a multi-stage OCR–LLM–Human pipeline that goes far beyond extracting OCR boxes. As detailed in the paper, this pipeline is non-trivial and is essential for producing reliable grounding annotations across heterogeneous text-rich images. It is not a simple “dataset reorganization” procedure.
>
> In addition to the benchmark, we also introduce technical approaches specifically designed for grounding on TRIG, including (a) an instruction-tuning framework adapted to OCR-free grounding, and (b) an embedding-based grounding module inspired by late-interaction architectures. These methods provide concrete improvements and insights under the new benchmark. They are included not as claims of universal superiority, but as meaningful baselines that advance grounding performance within the TRIG setting.
>
>
>
> ---
> >W2: Table 1 lacks comparisons with several quite related baseline methods, such as [1], [2] and [3], which weakens the empirical evaluation and makes it difficult to assess the claimed effectiveness of the proposed approach.
> [1] Seed1.5-VL Technical Report. Arxiv 2025.
> [2] CogAgent: A Visual Language Model for GUI Agents. CVPR 2024.
> [3] Harnessing Webpage UIs for Text-Rich Visual Understanding. ICLR 2025.
>
> We sincerely thank the reviewer for the suggestions. We have examined the three referenced works. Although they all involve visually complex screens, Seed1.5-VL, CogAgent, and UI-VLU focus primarily on GUI or webpage UI understanding, rather than grounding answer-supporting regions in unstructured text-rich images. Seed1.5-VL [1] is a general multimodal foundation model without a grounding module; CogAgent [2] targets GUI navigation and interaction rather than spatial evidence localization; and MultiUI [3] is centered on structured webpage comprehension instead of OCR-free grounding. These methods differ fundamentally in task objectives, input domains, supervision signals, and output formats, and none of them perform bounding-box grounding comparable to the setting we study. In the revised manuscript, we will expand the Related Work section to clearly articulate how these GUI-focused approaches differ from our grounding problem and why they are not directly comparable.
>
> Reference
> [1] Seed1.5-VL Technical Report. Arxiv 2025.
> [2] CogAgent: A Visual Language Model for GUI Agents. CVPR 2024.
> [3] Harnessing Webpage UIs for Text-Rich Visual Understanding. ICLR 2025.

---

### Official Review · Reviewer_rDRd · 2025-11-01

**Soundness:** 3
**Presentation:** 3
**Contribution:** 3
**Rating:** 4
**Confidence:** 4

**Summary:**

This paper introduces the TRIG (Text-Rich Image Grounding) task, which aims to evaluate and enhance the visual text localisation capabilities of MLLMs in text-rich document images. The authors have constructed the TRIG-Bench benchmark, consisting of 800 manually annotated samples, along with a synthetic training set comprising 90,000 samples, which was created using an OCR–LLM–human interaction process for data construction. Additionally, the paper proposes two baseline methods: a general instruction fine-tuning approach and an efficient embedded plug-and-play method. A systematic evaluation of multiple existing MLLMs is conducted, revealing significant limitations of current models in terms of complex document understanding and localisation.

**Strengths:**

1.	TRIG addresses text-intensive document scenarios, such as tables and infographics, which have been overlooked by existing visual localisation research. This work fills the gap in the verifiability and credibility assessment of MLLM in document question answering.
2.	By integrating OCR with GPT-4o for automated generation and manual verification, TRIG balances scale and quality. It particularly excels in annotating bounding boxes supporting answers with clearly defined semantic alignment objectives.
3.	Through metrics such as the instruction-following rate, TRIG uncovers fundamental deficiencies in existing models for complex tasks, highlighting weaker instruction comprehension in comparison to spatial reasoning. This provides critical insights for the research community.

**Weaknesses:**

1. In Table 1, several open-source models (such as LLaVA and Qwen-VL) exhibit an IoU of 0.00 under the OCR-free setting, which is counterintuitive. It is advisable to verify the experimental setup and evaluation code (including prompt templates, model versions, random seeds, input preprocessing, and output parsing scripts) and to include several typical failure examples to identify the source of the issue.
2. The description of the embedding method lacks clarity. Please explicitly state which similarity measure is employed, how the embeddings are derived, how the similarity matrix is constructed and normalised, and how the similarity matrix is summarised into a similarity vector. It is recommended to provide relevant formulas or pseudo-code, and to report key hyperparameters as well as computational complexity.
3. Although the proposed embedding method demonstrates advantages in terms of efficiency, it has not been adequately compared with recent related work under identical experimental conditions, which diminishes the credibility of the method's innovation. It is recommended to perform the comparison using a unified dataset, the same OCR input, and consistent evaluation metrics, and to present a comparison table detailing accuracy, inference delay, and memory usage.
4. The notation is incomplete: the symbol l  appears in equations (1) and (2) but is not defined.

**Questions:**

1. In Table 1, several open-source models, including LLaVA and Qwen-VL, exhibit an IoU of 0.00 in the OCR-free scenario, which is markedly inconsistent with their performance in the standard VQA task. This discrepancy could potentially stem from errors in the evaluation code, deficiencies in the prompt design, or issues with the parsing of model outputs. If it can be demonstrated that the evaluation is indeed accurate and that the model is incapable of generating valid coordinates, the current conclusion may be upheld. Conversely, if the issue pertains to the evaluation or the parsing format, a revision of the conclusion will be necessary.
2. The current description lacks sufficient detail. For instance, the method used for similarity measurement has not been addressed. It is advisable to include relevant formulas or pseudocode to elucidate the computational process leading from image and text embeddings to the resultant similarity vector. Additionally, the method of similarity measurement and the aggregation strategy should be thoroughly clarified.
3. The paper should be subjected to a systematic comparison with recent works in the public domain under identical conditions. The absence of a consistent comparative framework undermines the persuasiveness of the contributions made by this method.

---

> ### Author Response · Authors · 2025-11-22
>
> We sincerely appreciate the time and effort the reviewers took to evaluate our manuscript and provide valuable feedback.
>
> ---
>
> > W1: In Table 1, several open-source models (such as LLaVA and Qwen-VL) exhibit an IoU of 0.00 under the OCR-free setting, which is counterintuitive. It is advisable to verify the experimental setup and evaluation code (including prompt templates, model versions, random seeds, input preprocessing, and output parsing scripts) and to include several typical failure examples to identify the source of the issue.
>
>
> We sincerely thank the reviewer for the insight.  We have thoroughly examined our evaluation setting to ensure the correctness of our experiments. Here are some analyses on why the values are so small.
>
> Grounding requires a model not only to understand the query but also to precisely localize the supporting visual evidence by generating structured bounding-box coordinates. In the OCR-free setting, the model must infer the region solely from raw pixels and produce a valid box entirely from scratch. Moreover, compared with most of the object-level grounding tasks, the bounding boxes of texts are so small, making the task substantially more challenging. As shown in Table 3, the average area of the GT area is even less than 1%, which means that incorrectly predicting several pixels will make the IoU degrade to zero.
>
> Consistent with this, our instruction-following analysis (Table 5) shows that several models do not output any valid bounding box for most samples, either ignoring the localization instruction or producing malformed coordinate strings. In such cases, an IoU of 0 is the inevitable and correct outcome of the evaluation pipeline.
>
> Furthermore, we have re-verified all components of the setup, including prompts, seeds, preprocessing, model versions, and parsing scripts, and confirmed that the results are stable and reproducible. Moreover, recent studies [1,2,3] have reported the same phenomenon: open-source MLLMs frequently struggle with structured localization, often yielding missing or incorrect bounding boxes and exhibiting sharp performance drops under stricter IoU thresholds. These findings align closely with our observations.
>
>
> ---
>
> > W2: The description of the embedding method lacks clarity. Please explicitly state which similarity measure is employed, how the embeddings are derived, how the similarity matrix is constructed and normalised, and how the similarity matrix is summarised into a similarity vector. It is recommended to provide relevant formulas or pseudo-code, and to report key hyperparameters as well as computational complexity.
>
> We sincerely thank the reviewer for the comments. To answer your question specifically:
> 1. Which similarity measure is employed: We utilize the dot product, which can be found in line 310.
> 2. How the embeddings are derived: We follow the same method as ColPali [7], directly getting the last hidden states from the language model outputs, and separating them as text or visual embeddings by the token indices.
> 3. How the similarity matrix is constructed and normalised, and how the similarity matrix is summarised into a similarity vector: The equation for the similarity vector is shown in equation 6, where each item in S represents the averaged similarity between each visual embedding and all text embeddings. No further normalization is conducted.
> We apologize for the inconvenience. We will additionally include a pseudo-code in the appendix of the future revised version. Please let us know if you have any further questions.

---

> > ### Author Response · Authors · 2025-11-22
> >
> > > W3: Although the proposed embedding method demonstrates advantages in terms of efficiency, it has not been adequately compared with recent related work under identical experimental conditions, which diminishes the credibility of the method's innovation. It is recommended to perform the comparison using a unified dataset, the same OCR input, and consistent evaluation metrics, and to present a comparison table detailing accuracy, inference delay, and memory usage.
> >
> > We sincerely thank the reviewer for the insights.  We would like to clarify that the efficiency discussion in our paper refers specifically to the comparison between our embedding-based method and the instruction-tuning baseline under the TRIG setting. Inspired by ColPali, our embedding approach is designed as a lightweight extension within the instruction-tuning framework, rather than as a standalone, highly optimized grounding model. Its purpose is to provide an efficient plug-in alternative to autoregressive decoding when performing grounding on TRIG, enabling a fair comparison of inference latency within our two proposed paradigms, rather than claiming broader efficiency superiority.
> >
> > Recent grounding approaches such as PixelLM[4], COPL[5], and SimVG[6] follow very different design goals and rely on specialized training pipelines or supervision. Without substantial task-specific adaptation or retraining, these models perform poorly on text-rich image grounding, as reported in previous studies[1,2,3]. Therefore, comparing inference efficiency across such heterogeneous and specially-trained systems would not yield meaningful conclusions and would not reflect the intent of our embedding module.
> >
> > However, we will add a more detailed Related Work section in the revised manuscript to clarify the differences in purpose and design philosophy between our embedding method and these approaches, and to make explicit that our efficiency comparison is scoped to the TRIG setting.
> >
> > ---
> >
> > > W4: The notation is incomplete: the symbol l appears in equations (1) and (2) but is not defined.
> >
> > We sincerely thank the reviewer for pointing this out. The symbol l in equations (1) and (2) represents the response length, which is used for calculating the losses on the outputs.
> >
> >
> > Reference:
> > [1] Jiang, Y., Yan, X., Ji, G. P., Fu, K., Sun, M., Xiong, H., ... & Khan, F. S. (2024). Effectiveness assessment of recent large vision-language models. Visual Intelligence, 2(1), 17.
> > [2] Wei, F., Zhao, J., Yan, K., Zhang, H., & Xu, C. (2024). A large-scale human-centric benchmark for referring expression comprehension in the LMM era. Advances in Neural Information Processing Systems, 37, 69566-69587.
> > [3] Chen, J., Wei, F., Zhao, J., Song, S., Wu, B., Peng, Z., ... & Zhang, H. (2025). Revisiting referring expression comprehension evaluation in the era of large multimodal models. In Proceedings of the Computer Vision and Pattern Recognition Conference (pp. 513-524).
> > [4] Ren, Z., Huang, Z., Wei, Y., Zhao, Y., Fu, D., Feng, J., & Jin, X. (2024). Pixellm: Pixel reasoning with large multimodal model. In Proceedings of the IEEE/CVF Conference on Computer Vision and Pattern Recognition (pp. 26374-26383).
> > [5] Dai, M., Yang, L., Xu, Y., Feng, Z., & Yang, W. (2024). Simvg: A simple framework for visual grounding with decoupled multi-modal fusion. Advances in neural information processing systems, 37, 121670-121698.
> > [6] Jiang, H., & Lu, Z. (2024, September). Visual grounding for object-level generalization in reinforcement learning. In European Conference on Computer Vision (pp. 55-72). Cham: Springer Nature Switzerland.
> > [7] Faysse, M., Sibille, H., Wu, T., Omrani, B., Viaud, G., Hudelot, C., & Colombo, P. (2024). Colpali: Efficient document retrieval with vision language models. arXiv preprint arXiv:2407.01449.

---

### Meta-Review · Area_Chair_UG9m · 2026-01-06

**Summary:**

The paper introduces TRIG, a task and benchmark designed to evaluate and improve the visual text grounding capabilities of MLLMs for text-rich scenarios. The core contributions is the proposed TRIG-Bench (800 manually annotated samples), and a synthetic training set of 90k samples created, as well as  two baseline methods to evaluate the benchmark.

**Reviewer Concerns:**

The reviews are mixed. While reviewers agree that the problem is important and the data pipeline is well-designed, there are significant concerns regarding novelty, overlap with existing document grounding works, missing baselines, and potential experimental errors specifically the suspiciously low performance of open-source models.

**Reviewer Scores:**

All reviewers hold the scores after the discussion. The core issues are novelty and overlap w.r.t. previous benchmark works.

---

### Decision · Program_Chairs · 2026-01-26

Reject